# ROS amplification drives mouse spermatogonial stem cell self-renewal

Hiroko Morimoto[1], Mito Kanastu-Shinohara[1,2], Narumi Ogonuki[3], Satoshi Kamimura[3], Atsuo Ogura[3], Chihiro Yabe-Nishimura[4], Yoshifumi Mori[1], Takeshi Morimoto[5], Satoshi Watanabe[1], Kinya Otsu[6], Takuya Yamamoto[2,7,8], Takashi Shinohara[1]

Reactive oxygen species (ROS) play critical roles in self-renewal division for various stem cell types. However, it remains unclear how ROS signals are integrated with self-renewal machinery. Here, we report that the MAPK14/MAPK7/BCL6B pathway creates a positive feedback loop to drive spermatogonial stem cell (SSC) self-renewal via ROS amplification. The activation of MAPK14 induced MAPK7 phosphorylation in cultured SSCs, and targeted deletion of *Mapk14* or *Mapk7* resulted in significant SSC deficiency after spermatogonial transplantation. The activation of this signaling pathway not only induced *Nox1* but also increased ROS levels. Chemical screening of MAPK7 targets revealed many ROS-dependent spermatogonial transcription factors, of which BCL6B was found to initiate ROS production by increasing *Nox1* expression via ETV5-induced nuclear translocation. Because hydrogen peroxide or *Nox1* transfection also induced BCL6B nuclear translocation, our results suggest that BCL6B initiates and amplifies ROS signals to activate ROS-dependent spermatogonial transcription factors by forming a positive feedback loop.

## Introduction

Spermatogonial stem cells (SSCs) undergo continuous self-renewal and produce numerous progenitors that ultimately give rise to spermatozoa (Meistrich & van Beek, 1993; de Rooij & Russel, 2000). Although the frequency of SSCs in the testis is very low (0.02–0.03%) (Meistrich & van Beek, 1993; Tegelenbosch & de Rooij, 1993), these cells produce sperm throughout the life span of male animals. SSCs have a unique mode of self-renewal because they do not undergo asymmetric division; a single SSC produces two stem cells by self-renewal division or two differentiated cells by differentiating division. These two types of divisions are considered to occur at the same frequency to maintain a constant population size (Meistrich & van Beek, 1993; de

Rooij & Russel, 2000). Because excessive self-renewal division leads to the accumulation of SSCs and increased differentiating division depletes SSCs, imbalances between the two types of divisions can result in male infertility. Therefore, the regulation of these two types of divisions in SSCs requires sophisticated control, but the molecular factors that regulate self-renewal division remain largely unknown.

Studies within the last decade suggest that reactive oxygen species (ROS) influence various stem cells. For example, hematopoietic stem cells are sensitive to ROS, and increased ROS levels induce senescence and compromise stem cell function (Ito et al, 2006). Embryonic stem (ES) cells are sensitive to hydrogen peroxide–induced apoptosis but are resistant to oxidative stress–induced senescence, entering a transient cell cycle arrest state (Guo et al, 2010). However, ROS are not necessarily harmful for self-renewal because proliferative neural stem cells (NSCs) have high endogenous ROS levels (Le Belle et al, 2011). Likewise, transient generation of ROS activates hair follicle stem cells, thereby promoting hair growth, and accelerates burn healing (Carrasco et al, 2015). Thus, ROS can also promote self-renewal in some tissues. Whereas ROS-induced senescence and damage have been well characterized, little is known about how ROS promote self-renewal machinery.

ROS have important influences on SSCs. We recently found that constitutive active *Hras* transfection induces SSC self-renewal without the need for self-renewal factors as well as increases ROS (Morimoto et al, 2013). The addition of ROS inhibitors suppressed self-renewal division, whereas hydrogen peroxide increased cell recovery. These results suggest that self-renewal division is positively regulated by ROS in SSCs. Consistent with this notion, testes of mice deficient in *Nox1*, which generates ROS, are smaller than those of WT mice, and SSCs in *Nox1* KO mice have reduced self-renewal activity upon serial transplantation. Depletion of *Nox1* in vitro by shRNA significantly suppressed self-renewal. These results suggest that ROS generated by *Nox1* are necessary for self-renewal. This conclusion was unexpected because *Nox1* expression is relatively low in germ cells and

[1]Department of Molecular Genetics, Graduate School of Medicine, Kyoto University, Kyoto, Japan [2]Japan Agency for Medical Research and Development-Core Research for Evolutional Science, Tokyo, Japan [3]Institute for Physical and Chemical Research (RIKEN), Bioresource Center, Tsukuba, Japan [4]Department of Pharmacology, Kyoto Prefectural University of Medicine, Kyoto, Japan [5]Department of Clinical Epidemiology, Hyogo College of Medicine, Nishinomiya, Hyogo, Japan [6]Cardiovascular Division, King's College London British Heart Foundation Centre of Research Excellence, London, UK [7]Department of Life Science Frontiers, Center for iPS Cell Research and Application (CiRA), Kyoto University, Kyoto, Japan [8]Institute for the Advanced Study of Human Biology (WPI-ASHBi), Kyoto University, Kyoto, Japan

Correspondence: tshinoha@virus.kyoto-u.ac.jp

ROS are thought to be harmful to spermatogenesis. In fact, ROS suppression is a commonly accepted treatment for male infertility.

Although these studies demonstrated the critical roles of ROS generated by *Nox* genes, they are only weakly expressed in germ cells, and the link between ROS generation and self-renewal has not been elucidated. SSC self-renewal is based on the complex interplay between stably expressed transcription factors and cytokine-induced transcriptional activators (Kanatsu-Shinohara & Shinohara, 2013). The p38 MAPK appears to be involved in this process because 1) the addition of self-renewal factors to cultured SSCs induces sustained phosphorylation of p38 MAPK and 2) inhibition of p38 MAPK by a chemical inhibitor SB203580 suppresses self-renewal and down-regulates *Nox1* (Morimoto et al, 2013). However, there are several genes responsible for p38 MAPK activity, and the critical target of p38 MAPK activity in SSCs has not been identified. Most importantly, it remains unclear how ROS integrate with the self-renewal machinery in general. In this respect, SSCs are uniquely favorable for research because both stem cell culture and transplantation techniques are available (Brinster & Zimmermann, 1994; Kanatsu-Shinohara et al, 2003). In this study, we aimed to identify the gene responsible for p38 MAPK activity and investigate its effect in SSCs. Using KO mice, we carried out a functional analysis of p38 MAPK and found that ROS create a positive feedback loop to sustain self-renewal by activating the MAPK14/MAPK7/BCL6B pathway. This process of ROS amplification by an SSC transcription factor may explain why ROS play important roles despite relatively low expression levels of *Nox1*.

## Results

### Essential role of Mapk14 in SSC self-renewal

FGF2 and GDNF play critical roles in deriving germline stem (GS) cells, which are cultured SSCs with enriched SSC activity (Kanatsu-Shinohara et al, 2003). Addition of FGF2 and GDNF to GS cells that had been starved for cytokines induced ROS (Fig S1A), which suggested close relationship between ROS and self-renewal. We previously showed that the p38 MAPK inhibitor SB203580 inhibits GS cell proliferation and reduces *Nox1* expression (Morimoto et al, 2013). Depletion of *Nox1* by shRNA suppressed ROS (Fig S1B). Although there are four p38 MAPK isoforms, it has not been determined which isoform is involved in GS cell proliferation. We carried out real-time PCR analysis and found that *Mapk14* is expressed in many organs, including testis (Fig S2), and that GS cells express *Mapk14* most strongly among the four p38 MAPK isoforms (Fig 1A). However, GS cells also expressed *Mapk11*. Because both MAPK11 and MAPK14 are sensitive to SB203580, we used another inhibitor VX-745, which is more specific to MAPK14. We added VX-745 to GS cells cultured in the presence of self-renewal factors. As expected, VX-745 suppressed GS cell proliferation in a dose-dependent manner (Fig 1B and C). Moreover, flow cytometric analysis showed that GS cells treated with VX-745 generated significantly lower levels of ROS in a similar manner to *Nox1* knockdown (KD) (Fig 1D and Table S1). MAPK14 activation level was closely

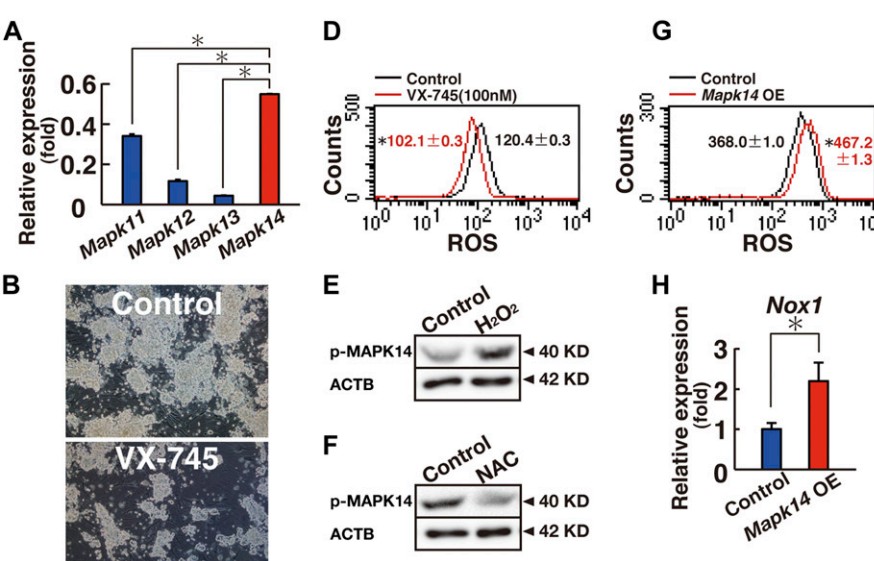

**Figure 1. MAPK14 regulates ROS generation in GS cells.**
**(A)** Real-time PCR analysis of *Mapk11*, *Mapk12*, *Mapk13*, and *Mapk14* in GS cells (n = 3 cultures). *$P < 0.05$ (ANOVA). **(B)** Appearance of GS cells after culturing with VX-745 for 6 d. Scale bar: 100 μm. **(C)** Cell recovery (n = 9 cultures). GS cells were cultured with VX-745 for 6 d. *$P < 0.05$ (ANOVA). **(D)** Flow cytometric analysis of ROS generation 5 d after VX-745 supplementation. (n = 3 cultures). *$P < 0.05$ (linear regression). **(E)** Western blot analysis of phosphorylated MAPK14 10 min after hydrogen peroxide (30 μM) supplementation. **(F)** Western blot analysis of phosphorylated MAPK14 2 d after NAC (0.5 mM) supplementation. **(G)** Flow cytometric analysis of ROS generation 1 d after transfection with constitutively active *Mapk14* (n = 3 cultures; multiplicity of infection [moi] = 4). *$P < 0.05$ (linear regression). **(H)** Real-time PCR analysis of *Nox1* expression 1 d after transfection with constitutively active *Mapk14* (moi = 4; n = 5 cultures). *$P < 0.05$ (*t* test). Data information: in (A, C, D, G, and H), data are presented as mean ± SEM.

related to ROS levels: although MAPK14 was phosphorylated by hydrogen peroxide treatment (Fig 1E and Table S2), addition of N-acetyl cysteine (NAC), a ROS scavenger, decreased MAPK14 phosphorylation (Fig 1F and Table S2). These results suggested that ROS generated by *Nox1* regulates MAPK14 and that MAPK14 plays some role in both GS cell proliferation and in ROS production.

MAPK14 phosphorylation is induced after cytokine supplementation (Morimoto et al, 2013). However, it was not clear whether MAPK14 plays any role in ROS generation. We transfected constitutively active *Mapk14* in GS cells to determine whether MAPK14 activation is sufficient for ROS generation. As expected, GS cells transfected with constitutively active *Mapk14* exhibited increased ROS generation by flow cytometry even in the presence of self-renewal factors (Fig 1G and Table S1). Because SB203580 inhibited *Nox1* expression in GS cells (Morimoto et al, 2013), we examined its expression level after transfection of constitutively active *Mapk14*. As expected, real-time PCR analysis showed up-regulation of *Nox1* after transfection (Fig 1H). These results showed a close correlation between MAPK14 activation and ROS.

We previously showed that the number of undifferentiated spermatogonia is significantly reduced in *Nox1* KO mouse testes (Morimoto et al, 2013). Using this KO mouse strain, we examined whether reduced ROS production can influence MAPK14 activity in vivo. To test this hypothesis, we performed immunohistochemistry for *Nox1* KO testes using antibodies against GFRA1 (a component of GDNF receptor; marker of $A_{single}$ [$A_s$], $A_{paired}$ [$A_{pr}$] spermatogonia, and some $A_{aligned}$ [$A_{al}$] spermatogonia), a component of GDNF receptor, to assess the phosphorylation levels of MAPK14 in these cells (Fig S3A) (Grasso et al, 2012). Enumeration of GFRA1+ cells positive for phosphorylated MAPK14 revealed that the number of GFRA1+ spermatogonia positive for phosphorylated MAPK14 was significantly reduced (Fig S3B), suggesting that NOX1-mediated ROS production is involved in MAPK14 phosphorylation in undifferentiated spermatogonia, possibly including SSCs. However, no changes were found in GATA4+ Sertoli cells, suggesting that ROS generation has a greater impact on germ cells (Fig S3C).

## Mapk14 gene targeting in SSCs

Based on these observations, we evaluated the function of MAPK14 by gene targeting. For this purpose, we used a floxed mutant mouse strain that possess loxP sites flanking exon 2 of the *Mapk14* gene (*Mapk14$^{f/f}$* mice) (Fig S4A). The cells were collected from 5- to 10-d-old pup testes, which are enriched with SSCs because of lack of differentiating germ cells. The cells were then exposed to a *Cre*-expressing adenovirus (AxCANCre) overnight (Takehashi et al, 2007). Testes from littermate mouse were used as controls. After trypsinization, an average of 67.1% and 59.0% of the input *Mapk14$^{f/f}$* and WT cells were collected, respectively, with no statistically significant difference (n = 7). Real-time PCR analysis confirmed the deletion of the target gene at the time of transplantation (Fig S4B).

Analysis of the recipients showed significantly reduced donor cell colonization after *Mapk14* deletion (Fig 2A). The numbers of colonies generated by *Mapk14* KO and control testis cells were 2.9 and 5.2 per 10$^5$ transplanted cells, respectively (Fig 2B). This difference was statistically significant. We also examined the colonization efficiency by counting the number of tubules exhibiting spermatogenesis by histology (Fig 2C). Whereas testes with mutant donor cells exhibited spermatogenesis in 3.6% of tubules, those that received control transplants exhibited spermatogenesis in 21.7% of tubules (n = 10). This difference was also statistically significant (Fig 2D).

To further investigate the role of MAPK14 in SSCs, we derived GS cells from control and *Mapk14$^{f/f}$* mice. Both types of GS cells were exposed to AxCANCre to remove the target gene. The virus-containing medium was removed the next day, and cells were recovered 6 d after culture initiation by trypsinization. Comparison of cell recovery showed that *Mapk14$^{f/f}$* GS cells proliferated significantly slower after AxCANCre exposure compared with control GS cells (Fig 2E). Moreover, these cells showed reduced ROS levels upon AxCANCre-mediated *Mapk14* deletion (Fig 2F and Table S1). As expected, *Mapk14* deletion resulted in reduced *Nox1* expression (Fig 2G). These results showed that activation of *Mapk14* is involved in GS cell proliferation.

## MAPK7 regulation by MAPK14

To further explore the signaling pathway involved in *Nox1* regulation, we screened GS cells for chemical activators/inhibitors that are involved in *Nox1* expression. We selected small chemicals that are involved in cell morphology (blebbistatin, pyrintegrin, thiazovivin, XMD 8-92, and Y27632), signal transduction (A83-010, BayK8644, Cx-4945, forskolin, Gö-6983, kenpaullone, MHY1485, NSC87877, PMA, quercetin, RepSox, and SC79), or epigenetic regulation (BIX01294, ML141, and NaB). Because many of these chemical inhibitors have known targets, screening would help identify candidate molecules that might regulate *Nox1* expression. Logarithmically growing GS cells were cultured for 3 d in the presence of candidate chemical compounds, and *Nox1* expression was analyzed by real-time PCR. Of the compounds evaluated, we found that XMD 8-92 efficiently reduced *Nox1* expression (Figs 3A and S5).

XMD 8-92 is a potent and selective inhibitor of MAPK7. GS cells treated with XMD 8-92 showed significantly reduced proliferation in a manner similar to that associated with MAPK14 inhibitors (Fig 3B). Western blot analysis showed that MAPK7 is activated upon cytokine stimulation (Fig 3C and Table S2). When GS cells were starved for 3 d and restimulated with self-renewal factors, MAPK7 phosphorylation was significantly increased, which suggests that MAPK7 plays a role in driving GS cell proliferation.

To determine whether MAPK7 is involved in ROS generation, we transfected GS cells with constitutively active *Map2k5*, which can activate MAPK7 (Terasawa et al, 2003). Real-time PCR analysis showed that transfection of constitutively active *Map2k5* induced *Nox1* expression (Fig 3D). In addition, flow cytometric analysis showed that exponentially growing GS cells produce a significant amount of ROS in constitutively active *Map2k5*-transfected cells (Fig 3E and Table S1). ROS generation by *Map2k5* also occurred in cells cultured without cytokines (Fig S1C). This was mediated by *Nox1* because *Nox1* suppression down-regulated ROS after *Map2k5* transfection (Fig S1D). The patterns of *Nox1* induction and ROS generation induced by *Mapk7* were similar to those induced by constitutively active *Mapk14*. These results suggest that MAPK7 is involved in the regulation of *Nox1* expression.

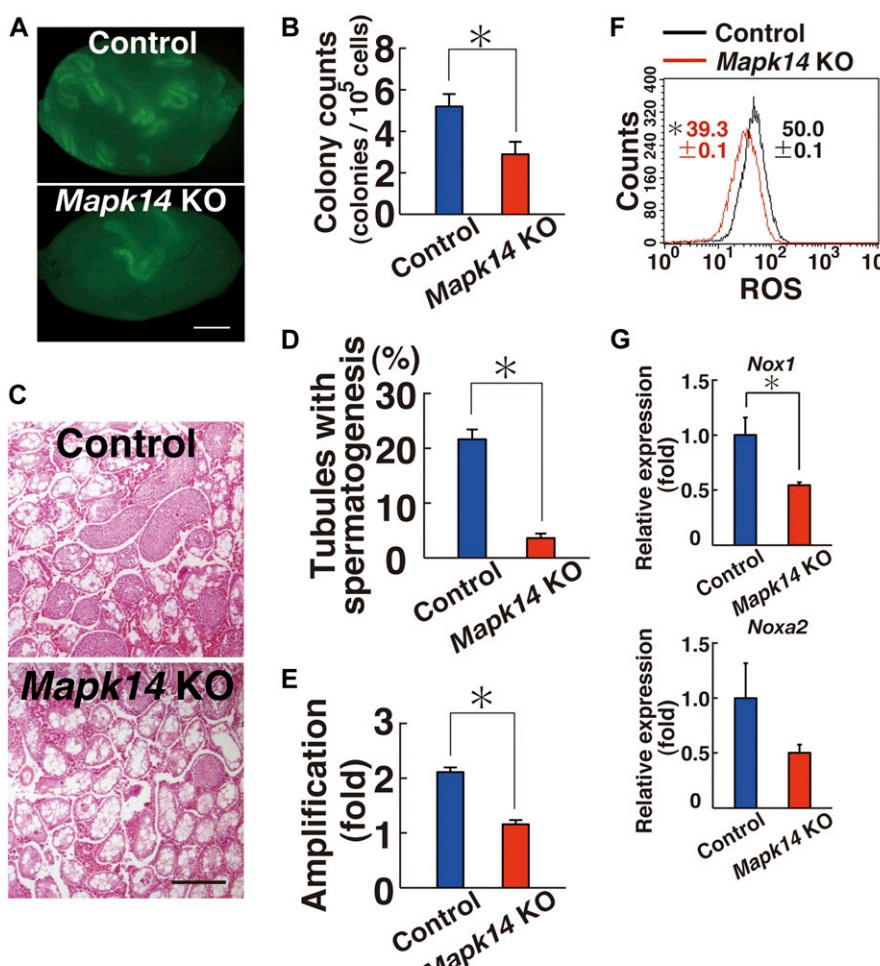

**Figure 2. Functional analysis of *Mapk14* in SSCs by spermatogonial transplantation.**
**(A)** Macroscopic appearance of a recipient testis that underwent transplantation of AxCANCre-treated *Mapk14^{f/f}* testis cells. Scale bar: 1 mm. **(B)** Colony counts (n = 25 testes for *Mapk14^{f/f}*; n = 20 testes for control). *P < 0.05 (*t* test). **(C)** Histological appearance of a recipient testis. Scale bar: 250 *μ*m. **(D)** Quantification of tubules exhibiting spermatogenesis. At least 1,139 tubules were counted. *P < 0.05 (*t* test). **(E)** Defective proliferation of *Mapk14^{f/f}* GS cells exposed to AxCANCre (n = 14 cultures; moi = 2). The cells were recovered 6 d after transfection. *P < 0.05 (*t* test). **(F)** Flow cytometric analysis of ROS generation in AxCANCre-treated *Mapk14^{f/f}* GS cells 1 d after transfection (moi = 2; n = 3 cultures). *P < 0.05 (linear regression). **(G)** Real-time PCR analysis of *Nox1* and *Noxa2* expression in *Mapk14^{f/f}* GS cells exposed to AxCANCre 4 d after infection (n = 3 cultures; moi = 2). *P < 0.05 (*t* test). Data information: in (B, D–G), data are presented as mean ± SEM.

We then examined the effect of ROS inhibitors on MAPK7 activation in stably growing GS cells. Western blot analysis showed that phosphorylation of MAPK7 is significantly suppressed by several *Nox* inhibitors, including diphenyleneiodonium (DPI), and apocynin (Fig 3F and Table S2). Similar results were found when we used α-lipoic acid, which is an antioxidant (Fig 3F). Because adding hydrogen peroxide increased MAPK7 phosphorylation (Fig 3G and Table S2), we examined the status of MAPK7 phosphorylation in vivo using *Nox1* KO mice (Morimoto et al, 2013). Immunohistochemistry of *Nox1* KO testes using antibodies against GFRA1 showed a significant reduction in MAPK7 phosphorylation in these cells, in a manner similar to MAPK14 (Fig S3D and E). These results suggest that ROS generation by *Nox1* induces MAPK7 and MAPK14 activation in GFRA1^+ undifferentiated spermatogonia in vivo.

Because MAPK7 and MAPK14 acted similarly on *Nox1* expression and ROS generation, we examined the relationship between these molecules. Because MAPK7 phosphorylation is mediated by its upstream kinase MAP2K5, we first confirmed the effect of MAP2K5 on MAPK7 by transfecting constitutively active *Map2k5*. Western blot analysis at 2 d after transfection showed increased levels of phosphorylated MAPK7, which suggests that MAPK7 is indeed activated by MAP2K5 (Fig 3H and Table S2). However, we could not

detect significant changes in MAPK14 phosphorylation levels in these cells. In contrast, when we transfected constitutively active *Mapk14*, we observed increased phosphorylation levels of both MAP2K5 and MAPK7 (Fig 3H). These results suggested that MAPK14 acts upstream of MAPK7 via MAP2K5.

To further confirm this relationship based on overexpression (OE) experiments, we used chemical inhibitors. SB203589 not only inhibited MAPK14 phosphorylation but also reduced MAPK7 phosphorylation, suggesting that MAPK14 activation is necessary for MAPK7 activation (Fig 3I and Table S2). In contrast, XMD 8-92 supplementation did not have any apparent effects on MAPK14 phosphorylation. The relationship between MAPK14 and MAPK7 was finally confirmed by *Mapk14^{f/f}* GS cells, which showed decreased phosphorylation of MAPK7 upon AxCANCre treatment (Fig 3J and Table S2). These results suggested that MAPK14 and MAPK7 phosphorylation levels are closely related to each other.

Although XMD 8-92 inhibits GS cell proliferation, it was possible that XMD 8-92 suppressed proliferation of progenitors and may not have effect on SSCs. This is because the frequency of SSCs is relatively low (~1–2%) in GS cell cultures (Kanatsu-Shinohara & Shinohara, 2013). Therefore, functional analysis of SSCs is required to demonstrate the effect of XMD 8-92 on GS cells. For this purpose, we

carried out germ cell transplantation. We transplanted the cells at the beginning and end of the culture period and measured the increase in SSC number. GS cells treated with XMD 8-92 stopped proliferating and generated a significantly smaller number of colonies after spermatogonial transplantation (Fig 3K and L). The total increase in SSC number was also significantly attenuated during culture (Fig 3M). These results confirmed that XMD 8-92 inhibits the self-renewal division of SSCs in GS cell cultures.

### Mapk7 gene targeting in SSCs

In the next set of experiments, we used a floxed mutant mouse strain that possess loxP sites flanking exon 4 of the *Mapk7* gene (*Mapk7*[f/f] floxed mice) to confirm the effects of XMD 8-92 (Fig S4C) (Wang et al, 2006). The cells were collected from 5- to 10-d-old pup testes, and exposed to AxCANCre overnight. Testes from littermate mouse were used as controls. The transfected cells were recovered for transplantation the next day. After transplantation, an average of 57.5% and 65.5% of the input *Mapk7*[f/f] and control cells were collected by trypsinization, respectively, and the difference was not statistically significant (n = 4). Real-time PCR analysis confirmed deletion of the target gene at the time of transplantation (Fig S4D). Three experiments were performed.

The analysis of recipient mice revealed significantly decreased colony numbers after CRE-mediated deletion of *Mapk7* (Fig 4A). Testis cells exposed to AxCANCre generated 4.7 colonies per 10[5] transplanted cells. In contrast, control testis cells generated 19.4

colonies per 10[5] cells (Fig 4B). The difference between the mutant and control samples was statistically significant. Histological analysis of the recipient testes also showed poor colonization of the *Mapk7*-deficient testis cells (Fig 4C). Spermatogenesis was detected in an average of 1.3% of the tubules that received mutant testis cells, whereas it was detected in an average of 6.7% of the seminiferous tubules that received control transplants (n = 10), and this difference was also statistically significant (Fig 4D).

To examine the role of MAPK7 in GS cell proliferation, we derived GS cells from *Mapk7* conditional KO mouse testes and exposed the cells to AxCANCre. In a manner similar to XMD 8-92, GS cells exposed to AxCANCre also proliferated slowly; whereas control cells increased 2.7-fold, *Mapk7* KO cells increased only 1.9-fold, and this difference was significant (Fig 4E). Moreover, ROS generation was significantly reduced when AxCANCre was added to *Mapk7*[f/f] GS cells (Fig 4F and Table S1).

Because only chemical inhibitors were used to study the mechanism of *Nox1* expression, we confirmed this relationship using *Mapk14*[f/f] and *Mapk7*[f/f] GS cells. As expected from the inhibitor experiments, real-time PCR showed down-regulation of *Nox1* expression levels in both GS cell types after treatment with AxCANCre (Figs 2G and 4G). AxCANCre treatment of *Mapk7*[f/f] GS cells also did not influence MAPK14 phosphorylation, which also confirmed that MAPK7 acts downstream of MAPK14 (Fig 4H and Table S2). Considering that MAPK7 phosphorylation is decreased in *Mapk14*[f/f] GS cells after AxCANCre treatment (Fig 3J), the results of these experiments showed that MAPK7 acts downstream of MAPK14.

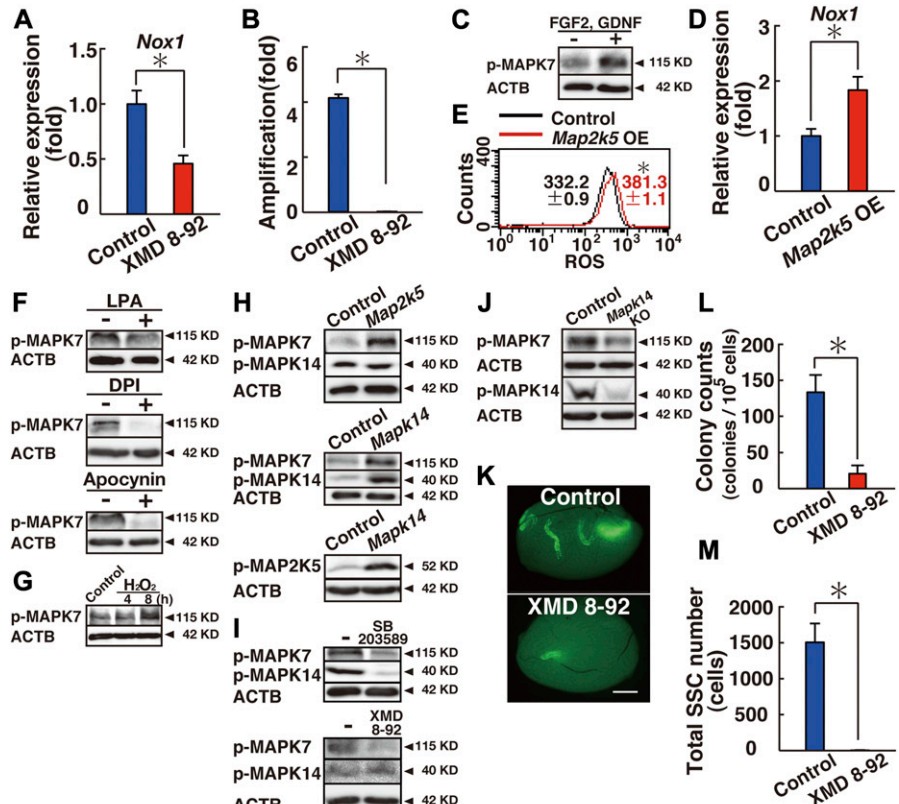

**Figure 3. Regulation of *Nox1* expression by MAPK7 in GS cells.**

**(A)** Real-time PCR analysis of *Nox1* expression 1 d after supplementation of XMD 8-92 (n = 6 cultures). *P < 0.05 (*t* test). **(B)** Defective proliferation of GS cells exposed to XMD 8-92 (n = 9 cultures). GS cells were cultured with XMD 8-92 for 5 d. *P < 0.05 (*t* test). **(C)** Western blot analysis of MAPK7 in GS cells treated with self-renewal factors. GS cells were cultured without cytokines for 3 d and restimulated with GDNF and FGF2 for 3 h. **(D)** Real-time PCR analysis of *Nox1* expression 1 d after transfection of constitutively active *Map2k5* (n = 6 cultures; moi = 4). *P < 0.05 (*t* test). **(E)** Flow cytometric analysis of ROS generation 2 d after transfection of constitutively active *Map2k5* (n = 3 cultures; moi = 4). *P < 0.05 (linear regression). **(F)** Western blot analysis of MAPK7 in GS cells 2 d after supplementation with the indicated ROS inhibitors. **(G)** Western blot analysis of MAPK7 phosphorylation. The cells were incubated with hydrogen peroxide (30 µM) for indicated time points before sample collection. **(H)** Western blot analysis of GS cells 2 d after transfection with constitutively active *Map2k5* or *Mapk14* (moi = 10). **(I)** Western blot analysis of GS cells 1 d or 4 d after supplementation with a MAPK14 or MAPK7 inhibitor, respectively. **(J)** Western blot analysis of *Mapk14*[f/f] GS cells 3 d after AxCANCre treatment (moi = 2). **(K)** Macroscopic appearance of a recipient testis that underwent transplantation of GS cells cultured with XMD 8-92 for 6 d. Scale bar: 1 mm. **(L)** Colony counts (n = 12 testes). *P < 0.05 (*t* test). **(M)** Total number of SSCs recovered from culture (n = 12 cultures). *P < 0.05 (*t* test). Data information: In (A, B, D, E, L, M), data are presented as mean ± SEM. *P < 0.05 (*t* test).

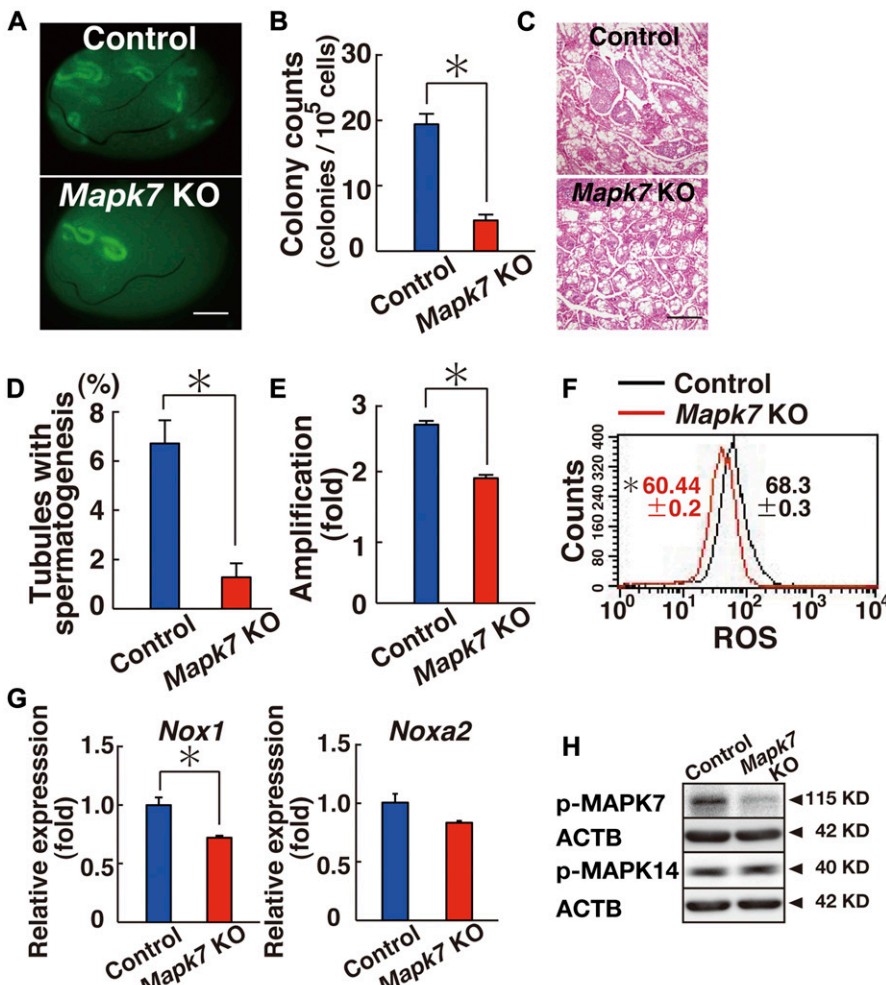

**Figure 4.  Functional analysis of *Mapk7* in SSCs using spermatogonial transplantation.**
**(A)** Macroscopic appearance of a recipient testis that underwent transplantation of AxCANCre-treated *Mapk7^{f/f}* testis cells. Scale bar: 1 mm. **(B)** Colony counts (n = 32 testes for *Mapk7*; n = 36 testes for control). *P < 0.05 (*t* test). **(C)** Histological appearance of a recipient testis. Scale bar: 250 *µ*m. **(D)** Quantification of tubules with spermatogenesis. At least 1,247 tubules were counted. *P < 0.05 (*t* test). **(E)** Defective proliferation of *Mapk7^{f/f}* KO GS cells exposed to AxCANCre (n = 16 cultures; moi = 2). The cells were recovered 6 d after transfection. *P < 0.05 (*t* test). **(F)** Flow cytometric analysis of ROS generation in AxCANCre-treated *Mapk7^{f/f}* GS cells 3–4 d after transfection (n = 3 cultures; moi = 2). *P < 0.05 (linear regression). **(G)** Real-time PCR analysis of *Nox1* and *Noxa2* expression in *Mapk7^{f/f}* GS cells exposed to AxCANCre 2 d after infection (n = 3 cultures; moi = 2). *P < 0.05 (*t* test). **(H)** Western blot analysis of *Mapk7^{f/f}* GS cells 2 d after AxCANCre treatment (moi = 2). Data information: in (B, D–G), data are presented as mean ± SEM.

## ROS generation by Mapk14 and Mapk7 activation

The generation of ROS by NOX1 requires additional components, including CYBA (p22phox), NOXO1 (or NOXO2 [p47phox]), NOXA1, and RAC1 (Bedard & Krause, 2007). Therefore, we searched for the genes responsible for ROS generation that allow ROS generation by co-transfection with *Nox1*. Real-time PCR analysis of stably growing GS cells showed that the cells express all of these genes except for *Noxo2* (Fig 5A). When we examined the cytokine response by adding GDNF and FGF2 to starved GS cells, the expression of *Noxa2* was most significantly up-regulated by cytokine restimulation (Fig 5B). This suggested that NOXA2 (p67phox) collaborates with NOX1 for ROS generation. In contrast, the addition of self-renewal factors only modestly changed the expression of other component genes. Therefore, these results suggested that *Noxa2* is an important regulator of ROS generation in GS cells and that *Nox1* and *Noxa2* are regulated in a similar manner.

To test this hypothesis, we first used chemical inhibitors. When GS cells are proliferating by GDNF and FGF2 stimulation, the expression of *Noxa2* was suppressed only by DPI (Fig 5C). Moreover, adding hydrogen peroxide induced the expression of *Nox1* but not *Noxa2* (Fig 5D). To confirm the impact of these genes, we checked its expression using *Mapk14* and *Mapk7* KO GS cells (Figs 2G and 4G). However, deletion of these genes did not show significant effects on *Noxa2* expression. These results suggested that *Nox1* and *Noxa2* are regulated in a different manner.

We also compared the effect of *Noxa2* in ROS generation and cell proliferation. Transfection experiments revealed that *Nox1* transfection alone is sufficient to generate ROS and enhances cell proliferation (Fig 5E and F, and Table S1). Although *Noxa2* depletion decreased ROS and impaired cell proliferation (Figs 5G, H, and S6A), neither ROS increase nor enhanced proliferation occurred after *Noxa2* transfection (Fig 5F and I, and Table S1). As expected, *Nox1* transfection generated ROS even when *Noxa2* was inhibited by shRNA (Fig 5J and Table S1). Spermatogonial transfection showed that the colony counts did not change after *Nox1* or *Noxa2* transfection (Fig 5K), which suggested that ROS levels did not influence SSC concentration in GS cell cultures. Taken together, these results suggest that NOX1 is primarily important for ROS generation.

## Involvement of Bcl6b in ROS generation in GS cells

Because the induction of ROS by *Nox1* transfection increased GS cell proliferation, we reasoned that a subset of transcription factors

acts downstream of ROS to drive self-renewal division. To identify such genes, we initially carried out microarray and RNA sequencing (RNA-seq) using *Mapk14*$^{f/f}$ GS cells, *Mapk7*$^{f/f}$ GS cells, or *Nox1* KD GS cells. However, we were not able to identify genes that are commonly involved in ROS regulation (Fig S7, Tables S3–S6). Because it was possible that lack of commonly expressed genes could be due to relatively low sensitivity of the assays (Mehta et al, 2016), we screened for spermatogonial transcription factor genes down-regulated by SB203589, XMD 8-92, and DPI by real-time PCR. Triplicate analysis of GS cells revealed down-regulation of many transcription factors whose deficiency causes defective spermatogenesis or self-renewal (Fig S8A), suggesting that ROS-dependent transcription factors have important functions in SSCs and spermatogenesis. In particular, the *Bcl6b* and *Sohlh1* genes were commonly down-regulated by the three inhibitors (Table S6).

To confirm this result, the effect of individual chemical inhibitors on these target genes was further examined by another set of inhibitors (BIRB796, BIX02189, and apocynin) against the same molecules. Although *Bcl6b* was down-regulated by all of these inhibitors, only apocynin down-regulated *Sohlh1* (Fig S8B). Moreover, when we checked the expression of all candidate genes in *Mapk14* KO, *Mapk7* KO, and *Nox1* KD GS cells by real-time PCR, only *Bcl6b* showed consistent down-regulation (Fig S8C). These results suggested that *Bcl6b* is the strongest candidate in ROS regulation.

To determine whether the candidate genes are involved in ROS generation, GS cells were transfected with shRNA and ROS levels were examined by flow cytometry 7 d after transfection. Whereas depletion of *Sohlh1* did not show reduced ROS, GS cells transfected with shRNA against *Bcl6b* showed significantly reduced ROS (Figs 6A and B, and Table S1). As expected, *Bcl6b* depletion also inhibited the ROS caused by transfection of constitutively active *Map2k5* (Fig 6B and Table S1). We then examined whether *Bcl6b* is sufficient for ROS generation and found that *Bcl6b* OE could indeed induced ROS (Fig 6B and Table S1). Consistent with this observation, real-time PCR analysis revealed that *Bcl6b* transfection induced *Nox1* in a dose-dependent manner (Fig 6C). Real-time PCR analysis also showed

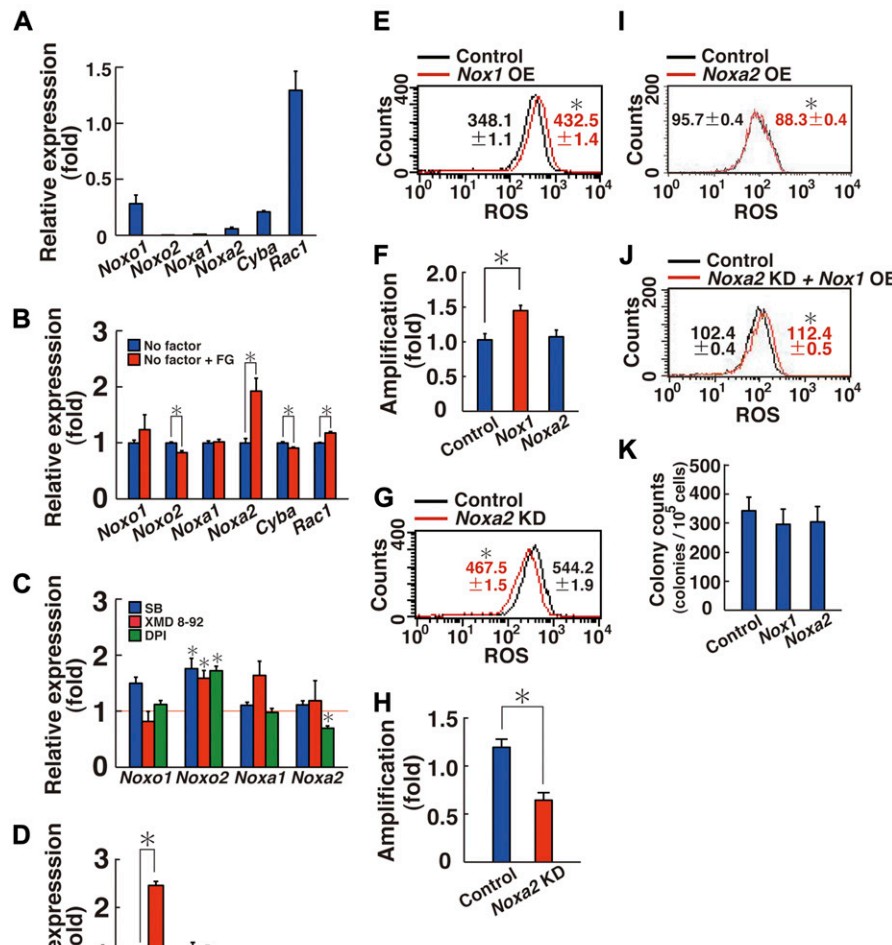

**Figure 5. ROS generation and enhanced proliferation of GS cells by *Nox1* transfection.**
**(A)** Real-time PCR analysis of *Nox1*-related components in GS cells (n = 3 cultures). **(B)** Real-time PCR analysis of *Nox1*-related components by FGF2 and GDNF (FG) supplementation (n = 3 cultures). The cells were starved for 3 d, and FGF2 and GDNF were added for 6 h before sample collection. *$P$ < 0.05 (*t* test). **(C)** Real-time PCR analysis of *Nox1*-related components 2 (DPI) or 3 d (SB203589 and XMD 8-92) after supplementation with the indicated inhibitors (n = 3 cultures). Mean values for control experiment are indicated by the horizontal line. *$P$ < 0.05 (*t* test). **(D)** Real-time PCR analysis of indicated genes 10 min after hydrogen peroxide (90 μM) supplementation (n = 3 cultures). *$P$ < 0.05 (*t* test). **(E)** Flow cytometric analysis of ROS generation 3 h after *Nox1* transfection (n = 3 cultures; moi = 10). **(F)** Cell recovery 5 d after *Nox1* or *Noxa2* transfection (n = 6 cultures; moi = 10). *$P$ < 0.05 (ANOVA). **(G)** Flow cytometric analysis of ROS generation 6 d after *Noxa2* depletion (n = 3 cultures; moi = 4). *$P$ < 0.05 (linear regression). **(H)** Cell recovery 7 d after *Noxa2* depletion (n = 6 cultures; moi = 4). *$P$ < 0.05 (*t* test). **(I)** Flow cytometric analysis of ROS generation 3 h after *Noxa2* transfection (n = 3 cultures; moi = 4). **(J)** Flow cytometric analysis of ROS generation 3 h after *Nox1* transfection into GS cells, in which *Noxa2* was depleted 6 d before (n = 3 cultures; moi = 4). *Eyfp* was transfected into *Noxa2*-depleted cells as a control. **(K)** Colony counts after spermatogonial transplantation of GS cells transfected with the indicated genes (n = 12 testes). *$P$ < 0.05 (*t* test). Data information: in (A–K), data are presented as mean ± SEM.

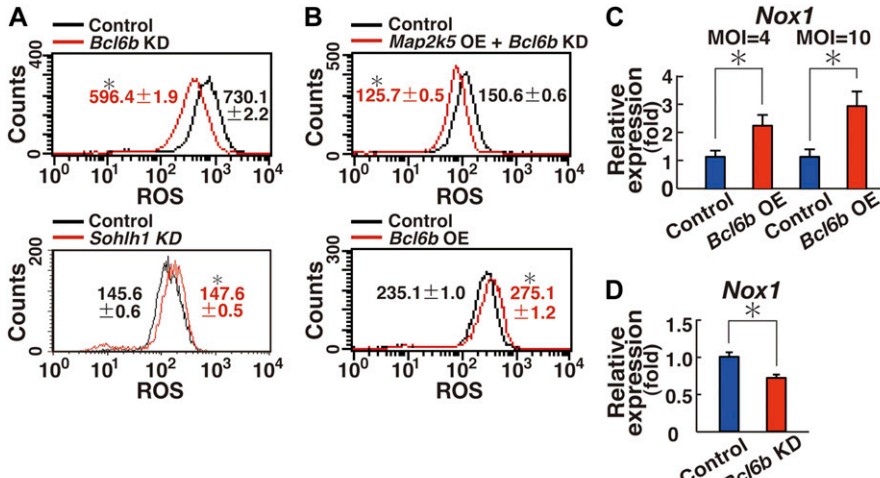

**Figure 6.  *Nox1* up-regulation by *Bcl6b*.**
**(A)** Flow cytometric analysis of ROS generation 7 d after depletion of the indicated genes by shRNA transfection (n = 3 cultures; moi = 4). *P < 0.05 (linear regression). **(B)** Flow cytometric analysis of ROS generation after transfection of the indicated genes (n = 3 cultures; moi = 4). The cells were recovered 2 (*Bcl6b* OE) or 4 d (*Map2k5* OE + *Bcl6b* KD) after transfection. *P < 0.05 (linear regression). **(C)** Real-time PCR analysis of *Nox1* expression 1 d after transfection with the indicated genes, respectively (n = 6 cultures for moi = 4; n = 4 cultures for moi = 10). *P < 0.05 (*t* test). **(D)** Real-time PCR analysis of *Nox1* expression 8 d after *Bcl6b* depletion (n = 6 cultures; moi = 4). *P < 0.05 (*t* test). Data information: in (A–D), data are presented as mean ± SEM.

*Bcl6b* KD significantly decreased *Nox1* expression (Fig 6D). These results suggest that *Bcl6b* is responsible for ROS generation.

### ETV5-mediated nuclear translocation of BCL6B

To clarify the relationship between BCL6B and the MAPK14/MAPK7 signaling pathway, we performed real-time PCR to examine changes in *Bcl6b* expression levels. Although GS cells transfected with constitutively active *Map2k5* showed a modest but significant increase in *Bcl6b* expression (Fig 7A), we reasoned that the modest induction of *Bcl6b* could not explain the significant increase in ROS generation. We then performed immunostaining of the transfected cells to assess the changes in BCL6B localization. Although BCL6B is a transcription factor, BCL6B was predominantly found in the cytoplasm of GS cells, whereas it exhibited only weak signals in the nucleus (Figs 7B and S9A). However, when GS cells were transfected with constitutively active *Map2k5*, the density of BCL6B increased by 2.5-fold in the nucleus (Fig 7B), which suggests that *Map2k5* regulates the nuclear translocation of BCL6B.

To gain insight into the underlying mechanism, we focused on the role of *Etv5*. *Etv5* is essential for SSC self-renewal, and *Etv5* KO mice results in loss of spermatogenesis (Chen et al, 2005). It also regulates CXCR4 expression and may regulate SSC homing (Wu et al, 2011). Although *Etv5* expression was not suppressed by XMD 8-92 or *Bcl6b* KD (Fig S8A and C), it was down-regulated in *Mapk14* KO GS cells (Fig S8C). Because *Etv5* transfection induces *Bcl6b* up-regulation in stable transfectants (Ishii et al, 2012), we hypothesized that ETV5 might play a role in BCL6B translocation into the nucleus. Consistent with this idea, when we depleted *Etv5* by shRNA, GS cells showed a significant decrease in ROS (Fig 7C and Table S1). The same treatment also blocked ROS induced by constitutively active *Map2k5* OE (Fig 7C). On the other hand, transfection of constitutively active *Etv5* generated ROS (Fig 7C), suggesting that ETV5 acts downstream of MAPK14/MAPK7 signaling pathway for ROS. As expected from these observations, transfection of constitutively active *Etv5* increased the number of cells expressing that contain BCL6B in the nucleus (Fig 7B). The density of BCL6B staining in the nucleus was increased 2.4-fold by constitutively active *Etv5*

transfection, suggesting that *Etv5* regulates BCL6B translocation into the nucleus. GS cells that had been depleted of *Etv5* or *Bcl6b* by shRNA showed significantly reduced SSC activity (Fig S10A and B), when constitutive active *Map2k5* was transfected, which confirmed the critical role of this signaling pathway in SSC self-renewal.

Because the addition of hydrogen peroxide promoted GS cell proliferation and increased *Nox1* expression (Fig 5D) (Morimoto et al, 2013), we reasoned that the ROS generated by *Nox1* also promote nuclear translocation of BCL6B, thereby creating a positive feedback loop. To test this hypothesis, we transfected *Nox1* or added hydrogen peroxide, both of which also increased BCL6B nuclear localization 2.7- or 2.1-fold, respectively (Figs 7B, D, and S9A). Although these experiments were performed using stably growing GS cells, ROS induction occurred even when *Bcl6b* was transfected into cytokine-starved GS cells (Fig 7C and Table S1), indicating that *Bcl6b* is primarily responsible for ROS generation even after cytokine deprivation. Although *Etv5* transfection induced modest increase in *Bcl6b* mRNA expression, none of the treatments that increase ROS levels did not influence protein levels of BCL6B (Fig S9B and C). Consistent with this idea, we also noted delayed phosphorylation of MAPK14 after *Map2k5* transfection (Fig 7E and Table S2). *Map2k5* transfection did not influence MAPK14 2 d after transfection, but it induced phosphorylation at 5–6 d after transfection. *Blc6b* expression was important for MAPK14 and MAPK7 activation because *Blc6b* KD down-regulated phosphorylated MAPK14 and MAPK7 levels 9 d after KD (Fig 7F and Table S2). Considering that MAPK14 activation induces MAPK7 phosphorylation and the fact that MAPK7 activation causes delayed MAPK14 activation (Figs 3H and 7E), these results suggested that a positive feedback loop operates between MAPK14 and MAPK7 (Fig 7I).

Finally, we examined the contribution of BCL6B to the defective growth of *Mapk14* KO and *Mapk7* KO GS cells. Immunostaining of both cell types showed that intranuclear BCL6B expression is decreased upon *Cre* treatment (Figs 7G, H, and S9A). Because *Cre* treatment of both cell types also decreased *Nox1* expression (Figs 2G and 4G), these results suggest that ETV5 drives the ROS generation by regulating the nuclear translocation of BCL6B, which subsequently up-regulates *Nox1* to induce ROS.

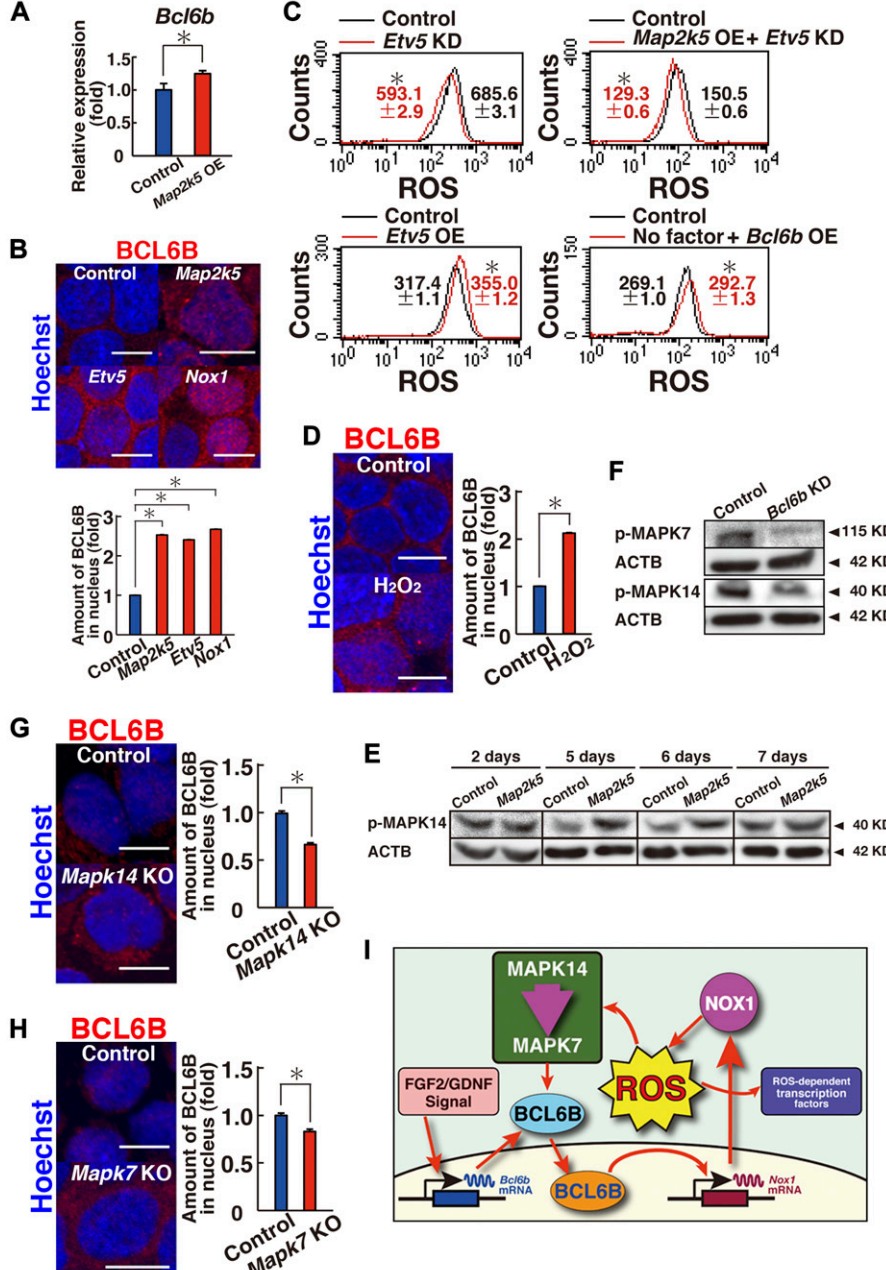

**Figure 7. Nuclear translocation of BCL6B.**
**(A)** Real-time PCR analysis of *Bcl6b* expression 1 d after transfection of *Map2k5* (n = 6 cultures; moi = 4). *$P$ < 0.05 (*t* test). **(B)** Immunostaining and quantification of nuclear BCL6B staining in GS cells 2 d after transfection of the indicated genes (moi = 4). At least 119 cells were counted for quantification. Scale bar: 5 $\mu$m. *$P$ < 0.05 (ANOVA). **(C)** Flow cytometric analysis of ROS generation following transfection of the indicated genes (n = 3; moi = 4). The cells were recovered on 3 (*Etv5* OE), 4 (*Map2k5* OE + *Etv5* KD), or 5 d (*Etv5* KD) after transfection. For cytokine starvation, the cells were cultured without cytokines for 2 d, and recovered 1 d after transfection. *$P$ < 0.05 (linear regression). **(D)** Immunostaining and quantification of nuclear BCL6B staining of GS cells 2 d after hydrogen peroxide treatment. At least 133 cells were counted for quantification. Scale bar: 5 $\mu$m. *$P$ < 0.05 (*t* test). **(E)** Western blot analysis of MAPK14 in GS cells at indicated time pointes after transfection with constitutively active *Map2k5* (moi = 4). **(F)** Western blot analysis of MAPK7 and MAPK14 in GS cells 9 d after *Bcl6b* KD (moi = 4). **(G, H)** Immunostaining and quantification of nuclear BCL6B staining of *Mapk14*^f/f (G) or *Mapk7*^f/f (H) GS cells 1 d after AxCANCre treatment (moi = 4). At least 205 cells were counted for quantification. Scale bar: 5 $\mu$m. *$P$ < 0.05 (*t* test). **(I)** Summary of results. MAPK14 induces MAPK7 phosphorylation, which promotes BCL6B entry into the nucleus via ETV5. BCL6B induces expression of *Nox1*. ROS generated by NOX1 activate MAPK14, thereby creating a positive feedback loop. ROS also activate many transcription factors. Data information: in (A–D, G, H), data are presented as mean ± SEM.

## Discussion

*Mapk14* has context-dependent roles in different tissues. It is involved in the proliferation of stem cells in the lungs and muscle (Ventura et al, 2007; Bernet et al, 2014), whereas its activation leads to senescence in hematopoietic stem cells (Ito et al, 2006). In this respect, SSCs are similar to stem cells in the lungs and muscle in that *Mapk14* positively contributes to self-renewal division. *Mapk14* activation can be caused by increased ROS levels. Previous studies have suggested a link between *Mapk14* and *Nox1*, but this relationship is not clear and appears to depend on the tissue type. In

the present study, we showed that ROS and MAPK14 create a positive feedback loop to promote SSC self-renewal. To our knowledge, this is the first demonstration of ROS-mediated positive feedback in sustaining stem cell self-renewal.

Our chemical screening identified MAPK7 as a critical regulator of ROS generation and *Nox1* expression. Although MAPK7 and MAPK1/3 share the Thr-Glu-Tyr activation motif, MAPK7 is different from MAPK1/3 in that MAPK7 has a long C-terminal domain with transcriptional activity. In addition, *Mapk7* KO mice show a phenotype distinct from that of *Mapk1/3* KO mice (Hayashi & Lee, 2004). Therefore, despite its similarity to MAPK1/3, MAPK7 plays a distinct

role. In mouse ES cells, MAPK7 phosphorylation occurred downstream of BMP4 via MAP2K5, which is critical for mediating BMP4-induced self-renewal (Morikawa et al, 2016). In GS cells, we found a critical role for MAPK7 in self-renewal, but it was apparently different because MAPK7 expression was accompanied by *Nox1* induction and ROS generation. Because *Mapk7* KO SSCs showed significantly reduced self-renewal activity and MAPK7 phosphorylation was induced by constitutively active *Mapk14* OE, we initially thought that MAPK7 acts downstream of MAPK14 and is similarly involved in ROS generation.

The importance of *Nox1* in ROS generation in GS cells was clearly demonstrated by transfection experiments. Although ROS can be induced in cultured cells by transfection of *Nox* genes, transfection of additional components, such as *Noxa* or *Noxo* genes is usually required (Bedard & Krause, 2007). In GS cells, however, many of these components are already expressed without induction by cytokines. Although self-renewal factor stimulation also increased *Noxa2* expression and *Noxa2* depletion decreased cell recovery, transfection of *Nox1* alone was sufficient for ROS production, suggesting that the level of *Nox1* expression is the most critical factor in ROS generation. Because *Nox1* OE increased ROS and adding hydrogen peroxide induced *Nox1* expression, these results led us to speculate that ROS-dependent transcription factors drive a positive feedback loop to sustain self-renewal.

Based on this notion, we screened transcription factors that respond to ROS signals and identified that *Bcl6b* is the most critical transcription factor for ROS generation. Although we failed to identify commonly regulated genes by microarray and RNA-seq, we think that this was caused by the low sensitivity of RNA-seq (Mehta et al, 2016). Moreover, the timing of RNA analysis is complicated by the positive feedback loop because each analyzed molecule contributes to the regulation of BCL6B localization in a different manner (i.e., phosphorylation, ROS generation, and transcription repression). The involvement of *Bcl6b* in SSC self-renewal was initially discovered by determining the genes induced by GDNF supplementation (Oatley et al, 2006). However, FGF2 also induces *Bcl6b* via *Etv5* induction (Ishii et al, 2012). Although *Bcl6b* was thought to drive self-renewal genes, our results suggest that the role of *Bcl6b* is not so simple. Our analyses suggest that *Nox1* is one of the critical effectors of BCL6B. Moreover, delayed MAPK14 phosphorylation by *Map2k5* transfection strongly suggested that ROS generated by NOX1 creates a positive feedback loop because ROS also induces BCL6B translocation and *Nox1* up-regulation. We think that this phase of self-renewal may be considered as maintenance phase, which should be analyzed separately from initial *Bcl6b* up-regulation by self-renewal signal. Therefore, BCL6B translocation is regulated by a complex manner involving MAPK14/MAPK7 signaling and ETV5. This link between BCL6B and NOX1 to amplify ROS may explain why *Nox1* plays such important roles in SSCs despite its relatively low expression levels.

At least three questions arose from this study. The first concerns the effect of ROS on transcription factors. Although only *Bcl6b* was able to induce ROS, several other transcription factors also showed significant changes in expression after ROS depletion by DPI and their significance in SSC self-renewal is unknown. Because chemicals can have off-targets, we still cannot totally exclude the possibility that other pathways may be involved. It has been known

that GDNF and FGF2 induce significant changes in the gene expression profiles. However, the degree and the impact of ROS-dependent and ROS-independent pathways on these changes in gene expression have been poorly addressed. The second question concerns understanding the mechanism of ETV5-mediated increase in nuclear BCL6B. We speculate that ETV5 may up-regulate molecules that help nuclear transport. BCL6, which is a close counterpart of BCL6B, interact with many proteins, including NCOR2 or BCOR (Huynh et al, 2000). It is possible that ETV5 may regulate such additional targets, which may help transport BCL6B into the nucleus. The last concerns the mechanism of ROS-based MAPK14 activation. ROS can activate MAPK by oxidative modification of MAPK signaling proteins (e.g., receptor tyrosine kinase or MAPK kinase; MKP) or inactivation of MAPK phosphatases. For example, ROS can activate the EGF receptor in the absence of its ligand. It has been also demonstrated that ROS-induced MKP inactivation causes sustained activation of JNK pathway (Hou et al, 2008). However, we still do not know such regulatory molecules responsible for feedback regulation in SSCs. Future studies are required to resolve these problems.

In many self-renewing tissues, ROS are harmful to stem cells, but some tissues require ROS for self-renewal. However, the positive role of ROS in self-renewal division has only recently begun to be analyzed. Our study shows that in SSCs, ROS amplification plays a critical role to drive self-renewal division. Because ROS can be toxic to germ cells and cause spermatogenic defect, success of spermatogenesis appears to depend on the delicate control of ROS levels. Identification and functional analysis of such ROS regulatory molecules will enhance our understanding of the molecular mechanisms of spermatogenesis and ROS-mediated self-renewal in tissue-specific stem cells.

# Materials and Methods

### Animals and cell culture

*Nox1* KO, *Mapk14^{f/f}*, and *Mapk7^{f/f}* conditional KO mice were generated as described previously (Nishida et al, 2004; Matsuno et al, 2005; Wang et al, 2006). These mice were crossed with transgenic Gt (ROSA)26Sor^{tm1(EYFP)Cos} mice (designated *R26R-Eyfp*; a gift from Dr. F Costantini, Columbia University Medical Center, New York, NY) to introduce a donor cell marker for transplantation experiments. GS cells from a transgenic mouse line of B6-TgR(ROSA26)26Sor mice (designated ROSA) (Jackson Laboratory) were previously described (Kanatsu-Shinohara et al, 2003), and GS cells were derived from *Mapk14^{f/f}* and *Mapk7^{f/f}* conditional KO mice using culture medium based on Iscove's modified MEM (Invitrogen), which was supplemented with 10 ng/ml human FGF2, 15 ng/ml recombinant rat GDNF (both from Peprotech), and 1% FBS, as described previously (Kanatsu-Shinohara et al, 2014). Cultures were maintained on mitomycin C–treated mouse embryonic fibroblasts (MEFs). Where indicated, hydrogen peroxide (30 $\mu$M) or NAC (0.5 mM; Sigma) was added to the cultures. The chemicals used in the study are listed in Table S7. FGF2 and GDNF are supplemented in the medium unless stated otherwise.

## Transplantation

Spermatogonial transplantation was carried out as described previously using busulfan (44 mg/ml)-treated C57BL/6 (B6) × DBA/2 F1 (BDF1) mice (Ogawa et al, 1997). Recipients were used at least 4 wk after busulfan treatment. For histological analysis to evaluate donor-derived spermatogenesis levels, we used 4–8-wk-old WBB6F1-W/Wᵛ (W) mice (Japan SLC) because they lack endogenous spermatogenesis. Approximately 10 or 4 $\mu$l of cell suspension was microinjected through the efferent duct of busulfan-treated or W mice, respectively. Each injection filled 75–85% of all seminiferous tubules. The Institutional Animal Care and Use Committee of Kyoto University approved all of our animal experimentation protocols.

## Viral transfection

For OE experiments, mouse *Mapk14* D176A-F327S (a gift from Dr. D Engelberg, National University of Singapore, Singapore) and mouse *Map2k5* cDNA (a gift from Dr. E Nishida, Kyoto University, Kyoto, Japan) were cloned into the CSII-EF-IRES2-Puro lentivirus vector. We also produced a mutant mouse *Etv5* (T135D/T139D/S142D) and introduced into CSII-EF-IRES2-Puro. CSII-EF-IRES2-Puro expressing *Bcl6b* was described previously (Ishii et al, 2012). CSII-EF-*Efyp*-IRES2-Puro was used as a control. For shRNA-mediated gene KD, all vectors were purchased from Open Biosystems. A mixture of lentiviral particles was used to transfect GS cells or testis cells. pLKO1-Scramble shRNA (Addgene) was used as a control (Open Biosystems). All KD vectors are listed in Table S8. Lentiviral transfection was carried out as described previously using polyethylenimine MAX (Polysciences) (Morimoto et al, 2015). The virus culture supernatant was concentrated by ultracentrifugation at 50,000 *g* for 2 h. The virus titer was determined by infection of 293T cells, using the qPCR Lentivirus titration (Titer) kit (Abcam). We used AxCANCre and AxCANLacZ (RIKEN BRC) for adenoviral infection. Virus particles preparation and titer determination were carried out as described previously (Takehashi et al, 2007).

## Analysis of recipient mice

Recipients were euthanized 2 mo after transplantation, and donor cell colonies were examined under UV light. Donor cell clusters were defined as colonies when the entire basal surface of the tubule was occupied and was at least 0.1 mm in length (Nagano et al, 1999).

For the evaluation of donor cell colonization levels by histology, W recipient testes were embedded in paraffin, and histological sections were stained with hematoxylin and eosin. The number of tubules with spermatogenesis, defined based on the presence of multiple layers of germ cells in the entire circumference of the tubules, was recorded for one section from each testis.

## Immunostaining

For immunohistochemistry of *Nox1* KO mouse testes, the samples were fixed in 4% paraformaldehyde for 2 h. They were then embedded in Tissue-Tek OCT Compound (Sakura Finetek) for cryosectioning. Immunostaining of cryosections was performed by treating samples with 0.1% Triton-X in PBS. For staining GS cells, single-cell suspensions were concentrated on glass slides by centrifugation using a Cytospin 4 unit (Thermo Electron Corp.). After immersing the slides in blocking buffer (0.1% Tween 20, 1% bovine serum albumin, and 10% donkey serum in PBS) for >1 h, the samples were incubated with antibodies against the indicated antigens. Secondary antibodies were incubated for 1 h at room temperature. The samples were counterstained with Hoechst 33342 (Sigma). The images were collected using confocal microscope (Fluoview FV1000D; Olympus). Images were analyzed by Photoshop software (Adobe Systems). Quantification of BCL6B levels was performed using MetaMorph software (Molecular Devices). All antibodies are listed in Table S9.

## Flow cytometry

To measure the ROS levels in GS cells, we used CellROX Deep Red Reagent (Thermo Fisher Scientific) following the manufacturer's instructions (Morimoto et al, 2015). In brief, the cells were dissociated with trypsin and incubated for 30 min at 37°C with 5 $\mu$M CellROX Deep Red in GS cell culture medium containing 1% FBS (Kanatsu-Shinohara et al, 2014). After washing with PBS containing 1% FBS (PBS/FBS), the stained cells were analyzed using the FACSCalibur (BD Biosciences). Three separate experiments using at least two different cell lines were carried out for each staining. Flow cytometric analysis of the same experimental data set was analyzed under the same conditions. All antibodies are listed in Table S9.

## Microarray

To examine genome-wide gene expression profiles, total RNA samples were extracted with TRIzol reagent (Invitrogen) and purified using the RNeasy cleanup system (QIAGEN). Microarray data collection and analysis were carried out as described previously (Tanaka et al, 2016).

## RNA-seq

For library preparation, total RNA was extracted using RNeasy Mini kit (QIAGEN) following the manufacturer's instructions. RNA-seq libraries were generated from 200 ng total RNA with the TruSeq Stranded mRNA LT Sample Prep kit according to the manufacturer's protocol (Illumina). The obtained RNA-seq libraries were sequenced on NextSeq 500 (86-bp single-end reads, Illumina).

The sequenced reads were mapped to the mm$^{10}$ mouse reference genome using HISAT2 (version 2.1.0) (Kim et al, 2015) with the GENCODE annotation gtf file (version M17) after trimming adaptor sequences and low-quality bases by cutadapt-1.16 (Martin, 2011). The uniquely mapped reads (mapping quality [MAPQ] ≥ 20) were used for further analyses. The differentially expressed genes (>2-fold changes and q-values <0.05 between each pair of samples) were identified using Cuffdiff within Cufflinks version 2.2.1 package and GENCODE annotation gtf file (version M17, protein coding genes) (Trapnell et al, 2013). Fragments per kilobase million (FPKM) were calculated as the expression values by Cuffdiff, and the low expressed genes (<10 FPKM) both in control and KD/KO samples were filtered out from the differentially expressed genes.

## PCR analyses

Total RNA was isolated using TRIzol (Invitrogen), and first-strand cDNA was synthesized using the Verso cDNA Synthesis kit (Thermo Fisher Scientific). For real-time PCR, the StepOnePlus Real-Time PCR System and *Power* SYBR Green PCR Master Mix were used according to the manufacturer's protocol (Applied Biosystems). Transcript levels were normalized relative to those of *Hprt*. PCR conditions were 95°C for 10 min, followed by 40 cycles of 95°C for 15 s, and 60°C for 1 min. Each reaction was performed in duplicate. PCR primer sequences are listed in Table S10.

## Western blot analysis

Samples were separated by SDS–PAGE, transferred onto Hybond P membranes (Amersham Biosciences), and incubated with primary antibodies. All antibodies are listed in Table S9.

## Statistical analyses and data visualization

Microsoft Excel was used for statistical evaluation using unpaired two-tailed *t* tests. Multiple comparison analyses were performed using ANOVA followed by Tukey's honest significant difference test. For analysis of flow cytometry, fluorescence values obtained from FlowJo software (TreeStar) were used to compare control and experimental samples by linear regression model adjusting for the timing of experiments. Adobe illustrator (CS5) was used for compiling figures and designing schematic figures. All variance is reported as SEM, and *P*-values for all statistically analyzed experiments are listed in Table S1.

## Data availability

The accession numbers for the microarray and RNA-seq data are GSE106103 and GSE118191, respectively.

# Supplementary Information

# Acknowledgements

We thank Ms. Y Ogata and S Ikeda for the technical assistance. This research was supported by Grants-in-aid for MEXT (18H05281, 25112003).

## Author Contributions

H Morimoto: conceptualization, formal analysis, investigation, and writing—original draft.
M Kanatsu-shinohara: conceptualization, investigation, and writing—original draft, review, and editing.
N Ogonuki: investigation.
S Kamimura: investigation.
A Ogura: investigation.
C Yabe-Nishimura: resources.
Y Mori: formal analysis.
T Morimoto: formal analysis.
S Watanbe: formal analysis.
K Otsu: resources.
T Yamamoto: data curation and investigation.
T Shinohara: conceptualization, formal analysis, funding acquisition, investigation, and writing—original draft, review, and editing.

## Conflict of Interest Statement

The authors declare that they have no conflict of interest.

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
