## [Reviewer comments · Life Science Alliance]

Life Science Alliance

ROS amplification drives mouse spermatogonial stem cell self-renewal

Hiroko Morimoto, Mito Kanatsu-Shinohara, Narumi Ogonuki, Satoshi Kamimura, Atsuo Ogura, Chihiro Yabe-Nishimura, Yoshifumi Mori, Takeshi Morimoto, Satoshi Watanbe, Kinya Otsu, and Takashi Shinohara

Corresponding author(s): Takashi Shinohara, Kyoto University

Review Timeline:	Submission Date:	2019-03-11
	Editorial Decision:	2019-03-11
	Revision Received:	2019-03-15
	Accepted:	2019-03-18

DOI: 10.26508/lsa.201900374

Scientific Editor: Andrea Leibfried

Report:

(Note: Letters and reports are not edited. The original formatting of letters and referee reports may not be reflected in this compilation.)

Please note that the manuscript was previously reviewed at another journal and the reports were taken into account in inviting a revision for publication at *Life Science Alliance* prior to submission to *Life Science Alliance*.

REFeree REPORTS OBTAINED DURING PEER REVIEW ELSEWHERE AND AUTHOR RESPONSES

Referee #1:

This manuscript presents the hypothesis that P38 MAPK regulates spermatogonial stem cell self-renewal through a complex feedback loop involving MAPK14, MAPK7, BCL6B and NADPH oxidase-generated ROS. While the experiments are generally strong and support the pathway, there are some general and specific concerns. General comments: The manuscript is long and very complicated. The Introduction is far too long and involved. The Results section has much that could be shortened and/or moved to Supplemental Files and the Discussion section spends a lot of time rehashing the Results. The authors should consider shortening the description of multiple experiments that address the same point, especially when they are in different locations.

Specific Comments:

1. The pathway is extraordinarily complicated. The manuscript would be greatly enhanced by the inclusion of a cartoon diagramming the proposed pathway.
2. On page 12 the authors mention that the MAPK7 inhibitor XMD 8-93 may affect progenitors but not stem cells. Therefore, to establish the effects on stem cells they used a functional analysis of SSCs (germ cell transplantation). However, other experiments were performed in the manuscript where conclusions were drawn about the effects on SSCs without doing this functional assay. This needs to be clarified. For example, on page 10-11 the authors describe stimulating GS cultures with cytokines that increase MAPK7 phosphorylation and "self-renewal" (proliferation) and conclude that this suggests MAPK7 plays a role in SSC self-renewal. However, this was just a proliferation assay and not the functional assay that the authors describe as a necessary test to determine stem cell-specific effects. The authors describe the cytokines used as "self-renewal factors". If there is some evidence that these cytokines induce effects only in the stem cell population of GS cultures they should add this data prior to this figure or refer to a prior publication that established this specialized stem cell effect. Otherwise, it seems that some experiments are described as finding stem cell specific effects without having performed the necessary functional assays to back up the claim.
3. Throughout the manuscript/experiments changes in cellular ROS levels established by FACS are shown as a single intensity plot from the FACS without any quantification. In many instances, the change in ROS levels appears to be very small though they are described as significant. The experiments should have been run more than once and the quantified data on the ROS levels, including the error bars indicating the variability of the results, should be shown along with statistics showing if the differences are significant. It is unclear from the manuscript how many times the FACS experiments were performed and on how many independent cell lines.

Referee #2:

Summary

In this follow-up study (from observations published in Morimoto et al., 2013 and Morimoto et al., 2015), the authors extend our understanding of the role of ROS in SSC self-renewal by further clarifying the relationship between ROS generation and SSC self-renewal; particularly focusing on the underlying mechanisms downstream of p38 MAPK. More interestingly, the authors propose a novel model implicating MAPK14 and ROS in a positive feedback loop that sustains SSC self-renewal.

Major Comments

Using a combination of loss-of-function (genetic ablation, and small molecule inhibitors) and gain-of-function (constitutively active mutants) assays, the authors generally provide conclusive evidence regarding the contribution of the MAPK14/MAPK7/BCL6B axis to NOX1-mediated ROS generation and SSC self-renewal. However, the authors should consider addressing the following:

1. Although the requirement for BCL6B and ETV5 in SSCs has been reported previously, the authors should confirm the role of Bcl6b and ETV5 within the context of MAPK14-mediated ROS generation. For instance, the authors may consider performing germ cell transplantation of GS cell cultures transfected with shRNA (against Bcl6b or Etv5) in the presence of ROS induction (via hydrogen peroxide or constitutively active Map2k5).

Unfortunately, evidence supporting the presence of a positive feedback loop is inconclusive, and more evidence addressing the following points needs to be provided:

2. Apart from Fig 1H, evidence of MAPK14 activation by ROS is limited. The authors should examine if (a) hydrogen peroxide can increase MAPK14 phosphorylation, and (b) if ROS scavengers can disrupt MAPK14 activation. In addition, it would be preferable to provide data (or some discussion) addressing how NOX1-derived ROS leads to MAPK14 activation.

3. The authors provide evidence (Fig. 1H) suggesting that "NOX1-mediated ROS production is involved in MAPK14 phosphorylation". However, constitutively active Map2k5, which leads to increased Nox1 expression and ROS levels (Fig 3D and E), does not increase MAPK14 phosphorylation (Fig 3H). Similarly, MAPK7 deletion does not reduce MAPK14 phosphorylation. If a positive feedback loop were involved, one would expect to observe altered MAPK14 phosphorylation following constitutively active Map2k5 expression or Mapk7 deletion; even if MAPK7 acts downstream of MAPK14. Notably, the kinetics of MAPK14 phosphorylation (resulting from MAPK7 activation/deletion) would also be expected to be delayed (relative to cytokine supplementation or ROS induction); since transcriptional upregulation of Nox1 via Bcl6b is implicated. As such, the authors should consider assaying the kinetics of MAPK14 phosphorylation (over several time intervals) in response to MAPK7 activation (i.e. constitutively active Map2k5).

Minor Comments

4. Regarding the importance of Noxa2 in Nox-1 mediated ROS generation, the authors conclude that "Nox1 expression is the most critical factor in ROS generation" based on the observation that Nox1 transfection alone generates ROS. However, it is possible that endogenous Noxa2 is present in amounts sufficient to support additional Nox1 activity. As such, the authors may consider including data on ROS generation after Nox1 transfection AND Noxa2 shRNA knockdown in Fig 5.

5. For all ROS histogram plots, the authors should indicate the MFIs (or fold-change) of the respective treatment conditions. In addition, it would be ideal if the MFIs from replicate experiments are provided in the form of a bar chart (or equivalent) next to the histogram plots. This will facilitate comparing the magnitude of change in ROS levels between various experiments (e.g. MAPK14 deletion vs MAPK7 deletion).

6. The authors should also provide ROS data for Nox-1 KO cells in the presence of cytokines or constitutively active Map2k5. This will serve as a reference for the magnitude of ROS generated by NOX1.

7. The authors should consider providing quantification (relative to reference gene) for Western blot results.

8. The authors may consider including an illustration of the proposed positive feedback model in the last Figure to facilitate comprehension of the study at a glance.

Referee #3:

In this study, Morimoto and the colleagues investigate the MAPK-dependent reactive oxygen species (ROS) amplification mechanism, which regulates the self-renewal of mouse spermatogonial stem cells (SSCs). By loss-of-function analysis, the authors showed that specific MAPK proteins are critical for the self-renewal of SSCs and the production of ROS in parallel, suggesting the presence of a feedback mechanism for the amplification of ROS. The following biochemical analysis identified the BCL6B as a mediator for ROS amplification downstream of MAPK signaling. In conclusion, these results suggest that the MAPK14/MAPK7/BCL6B pathway creates a positive feedback loop to promote the self-renewal of SSCs.

Major concerns:

The authors' finding showing the function of specific MAPKs in the self-renewal of SSCs and the production of ROS is novel. The detailed mechanistic analysis would potentially offer novel insights into the growth machinery of SSCs. However, some of the data presented in the manuscript seem to lack coherence and do not appropriately support the hypothesized ROS-amplification mechanism. For example, the authors performed microarray analysis using loss-of-function models and chemical screening analysis to identify the downstream mediators. However, the affected gene profile seems to be quite distinct between these 2 independent analyses. Actually, Bcl6b was identified only in the relatively small-scale chemical screening, suggesting that BCL6B is not a mediator downstream of MAPK14/7, as the authors claim. This inconsistency also suggests the possibility that BCL6B is one of the off-targets of chemical inhibitors.

Furthermore, throughout the manuscript, the presented results in RT-PCR and Western blotting lack the quantification data to support the authors' claim. Also, in the FACS analysis, it is difficult to estimate to what degree ROS production is affected by a specific condition in each experiment. For much of the data presented in the manuscript, the effects of a specific condition seem to be quite nominal. On the other hand, the signaling intensity measured by FACS analysis is quite changeable, even in the same biological condition as shown in the presented data (e.g. the signal intensity of the control shown in Fig. 6B, C), which suggests that there is considerable variation between experiments. Thus, the authors should present replicated data in each FACS experiment as described in the materials and methods. Moreover, additional measurements of ROS production by other methods would reinforce the authors' claim.

Finally, the authors claim that the positive feedback loop created by MAPK14/MPAK7/BCL6B signaling induces ROS amplification, which promotes SSC self-renewal. However, the presented results do not adequately support this scenario. To confirm the positive feedback loop, the authors should examine whether p-MAPK14 and p-MPAK7 are disrupted in Bcl6b KO SSC/GS cells by the feedback effect. Additionally, transcriptomic analysis of Bcl6b KO GS cells should be done to assess whether a subset of genes is commonly affected in Bcl6b KO GS cells, Mapk14f/f GS cells, Mapk7f/f GS cells, and Nox1 KD GS cells.

In conclusion, the current form of the manuscript does not meet the requirements for publication in the EMBO Journal.

Other specific comments:

Fig. 1A

The authors claim that "Mapk11, Mapk12, and Mapk14 are strongly expressed in GS cells" in the manuscript (page 7, line 5). However, the presented data do not support this notion. To clarify how strongly each Mapk gene is expressed in GSCs, the authors should quantify the expression level in several other tissues by real-time PCR.

Fig. 1D

The description of the FACS plot is confusing. The authors should describe which

colored FACS plot shows the specific condition as shown in Fig. 1E.

Fig. 1D, E

To confirm the quantification results by FACS, the authors should present at least two to four FACS replicates as described in the materials and methods.

Fig2.H

The description of the FACS plot is confusing. The authors should describe which colored FACS plot shows the specific condition as shown in Fig. 1E.

To confirm the quantification results by FACS, the authors should present at least two to four FACS replicates as described in the materials and methods.

Fig2.I

The authors should quantify the image of the Western blot bands. To confirm the regulation of Nox1 by Mapk14, they should also perform real-time PCR.

Fig.3E

The description of the FACS plot is confusing. The authors should describe which colored FACS plot shows the specific condition as shown in Fig. 1E.

To confirm the quantification results by FACS, the authors should present at least two to four FACS replicates as described in the materials and methods.

Fig.3F-J

The authors should quantify the band image of the Western analysis. The molecular weight of the specific protein band should be shown in each panel.

Fig.3H, I

The band pattern of p-MAPK14 seems to be different between each panel, which is confusing. The cutting of the images should be done in the same manner.

Fig.4

The authors should examine whether ROS production is affected in the Mapk7 KO GS cells as is the case in the Mapk14 KO GS cells.

Fig.4H

The quantification of the gene expression level should be done by real-time PCR.

Fig.5B, C

The authors should do the real-time PCR analysis to quantify the gene expression level.

Fig.5D, F

The description of the FACS plot is confusing. The authors should describe which colored FACS plot shows the specific condition as shown in Fig. 1E.

To confirm the quantification results by FACS, the authors should present at least two to four FACS replicates as described in the materials and methods.

Fig.5F

Here, the authors claim that Nox1 transfection alone is sufficient to generate ROS. To this reviewer, however, the knockdown experiment seems to suggest that Noxa2 plays a critical function in ROS production. As the following overexpression experiment is done in the presence of endogenous Noxa2, it is not clear how Nox1 alone is sufficient to generate ROS. To confirm their claim, the authors should assess whether Nox1 OE can rescue the reduction of ROS generation even in the Noxa2 KD condition. They should also overexpress Noxa2 and examine the effect on ROS production and SSC self-renewal.

Fig.5H

The authors should quantify both Noxa1 and Noxa2 levels by real-time PCR analysis.

Fig.6A

As described above, the presented data and related descriptions do not show the MAPK14/MAPK7/BCL6B pathway as the authors claim. The authors should explain the inconsistency between the microarray analysis and the chemical screening. They should also exclude the possibility that these results are caused by off-target effects. The quantification of gene expression should be done by real-time PCR.

Fig.6B, C, Appendix Fig S4

The knockdown efficiency of Bcl6b and other estimated factors should be confirmed by real-time PCR.

The description of the FACS plot is confusing. The authors should describe which colored FACS plot shows the specific condition as shown in Fig. 1E.

To confirm the quantification results by FACS, the authors should present at least two to four FACS replicates as described in the materials and methods.

Fig.7B, D

In Fig. 7A, the signal intensity of BCL6B in the overexpression conditions seems to be noticeably higher than that of the control, which is inconsistent with the modest activation of BCL6B by Map2k5 as shown in Fig 7A. In Fig 7D, however, the BCL6B signal of the control seems to be higher than the control signal in Fig. 7B, which is confusing. The authors should present the image in low-power field.

Additionally, to confirm the result, the authors should perform real-time PCR and Western blotting.

Fig.7C

The description of the FACS plot is confusing. The authors should describe which colored FACS plot shows the specific condition as shown in Fig. 1E.

To confirm the quantification results by FACS, the authors should present at least two to four FACS replicates as described in the materials and methods.

Fig.7E, F

The representative images seem to show that BCL6B is not expressed in Mapk14 KO and Mapk7 KO GS cells. However, the authors quantified the levels of BCL6B in the nucleus in Mapk14 KO and Mapk7 KO GS cells, which is confusing. The authors should present the image in low-power field.

Furthermore, to confirm the result, the authors should perform real-time PCR and Western blotting.

Referee #1:

Thank you for your positive comments on our manuscript. We appreciate your constructive comments on our manuscript.

This manuscript presents the hypothesis that P38 MAPK regulates spermatogonial stem cell self-renewal through a complex feedback loop involving MAPK14, MAPK7, BCL6B and NADPH oxidase-generated ROS. While the experiments are generally strong and support the pathway, there are some general and specific concerns.

General comments: The manuscript is long and very complicated. The Introduction is far too long and involved. The Results section has much that could be shortened and/or moved to Supplemental Files and the Discussion section spends a lot of time rehashing the Results. The authors should consider shortening the description of multiple experiments that address the same point, especially when they are in different locations.

With your suggestion, we shortened introduction and made changes in discussion section. We also moved some figures to extended view figure section and Appendix Figure section.

Specific Comments:

1. The pathway is extraordinarily complicated. The manuscript would be greatly enhanced by the inclusion of a cartoon diagramming the proposed pathway.

We agree. With your suggestion, we made a summary figure that incorporates our findings and current thoughts (Fig 7I).

2. On page 12 the authors mention that the MAPK7 inhibitor XMD 8-93 may affect progenitors but not stem cells. Therefore, to establish the effects on

stem cells they used a functional analysis of SSCs (germ cell transplantation). However, other experiments were performed in the manuscript where conclusions were drawn about the effects on SSCs without doing this functional assay. This needs to be clarified. For example, on page 10-11 the authors describe stimulating GS cultures with cytokines that increase MAPK7 phosphorylation and "self-renewal" (proliferation) and conclude that this suggests MAPK7 plays a role in SSC self-renewal. However, this was just a proliferation assay and not the functional assay that the authors describe as a necessary test to determine stem cell-specific effects. The authors describe the cytokines used as "self-renewal factors". If there is some evidence that these cytokines induce effects only in the stem cell population of GS cultures they should add this data prior to this figure or refer to a prior publication that established this specialized stem cell effect. Otherwise, it seems that some experiments are described as finding stem cell specific effects without having performed the necessary functional assays to back up the claim.

Thank you for pointing this out. We tried to be careful as much as possible, but this experiment escaped our attention. We modified the sentence to indicate that MAPK7 is involved in GS cell proliferation (page 10, line 9).

3. Throughout the manuscript/experiments changes in cellular ROS levels established by FACS are shown as a single intensity plot from the FACS without any quantification. In many instances, the change in ROS levels appears to be very small though they are described as significant. The experiments should have been run more than once and the quantified data on the ROS levels, including the error bars indicating the variability of the results, should be shown along with statistics showing if the differences are significant. It is unclear from the manuscript how many times the FACS experiments were performed and on how many independent cell lines.

With your suggestion, we indicated the number of experiments and quantified the ROS levels. Because of space limitation, we were not able to indicate error bars in the figure. Instead, SEMs are shown in the figures and all statistical analyses on FACS experiments were indicated in Appendix Table S1.

Thank you for your comments, particularly about the self-renewal division and summary figure.

Referee #2:

Thank you for your positive comments on our manuscript. Eight points were raised in the review.

Summary

In this follow-up study (from observations published in Morimoto et al., 2013 and Morimoto et al., 2015), the authors extend our understanding of the role of ROS in SSC self-renewal by further clarifying the relationship between ROS generation and SSC self-renewal; particularly focusing on the underlying mechanisms downstream of p38 MAPK. More interestingly, the authors propose a novel model implicating MAPK14 and ROS in a positive feedback loop that sustains SSC self-renewal.

Major Comments

Using a combination of loss-of-function (genetic ablation, and small molecule inhibitors) and gain-of-function (constitutively active mutants) assays, the authors generally provide conclusive evidence regarding the contribution of the MAPK14/MAPK7/BCL6B axis to NOX1-mediated ROS generation and SSC self-renewal. However, the authors should consider addressing the following:

- 1. Although the requirement for BCL6B and ETV5 in SSCs has been reported previously, the authors should confirm the role of Bcl6b and ETV5 within the context of MAPK14-mediated ROS generation. For**

instance, the authors may consider performing germ cell transplantation of GS cell cultures transfected with shRNA (against Bcl6b or Etv5) in the presence of ROS induction (via hydrogen peroxide or constitutively active Map2k5).

Unfortunately, evidence supporting the presence of a positive feedback loop is inconclusive, and more evidence addressing the following points needs to be provided:

We agree. We carried out transplantation experiments that you suggested. As we expected, the number of colonies was significantly decreased by Bcl6b or Etv5 KD even with Map2k5 overexpression. We included the results in result section (page 19, line 1) and included Fig EV5.

2. Apart from Fig 1H, evidence of MAPK14 activation by ROS is limited. The authors should examine if (a) hydrogen peroxide can increase MAPK14 phosphorylation, and (b) if ROS scavengers can disrupt MAPK14 activation. In addition, it would be preferable to provide data (or some discussion) addressing how NOX1-derived ROS leads to MAPK14 activation.

We did the experiments and found that hydrogen peroxide can increase MAPK14 phosphorylation and that N-acetylcysteine can decrease the MAPK14 activation (page 6, line 7 from the bottom)(Fig 1E, F). We also included several sentences in the discussion about how ROS leads to MAPK14 activation (page 23, line 4).

3. The authors provide evidence (Fig. 1H) suggesting that "NOX1-mediated ROS production is involved in MAPK14 phosphorylation". However, constitutively active Map2k5, which leads to increased Nox1 expression and ROS levels (Fig 3D and E), does not increase MAPK14 phosphorylation (Fig 3H). Similarly, MAPK7 deletion does not reduce MAPK14 phosphorylation. If a positive feedback loop were involved, one would expect to observe altered MAPK14 phosphorylation following

constitutively active Map2k5 expression or Mapk7 deletion; even if MAPK7 acts downstream of MAPK14. Notably, the kinetics of MAPK14 phosphorylation (resulting from MAPK7 activation/deletion) would also be expected to be delayed (relative to cytokine supplementation or ROS induction); since transcriptional upregulation of Nox1 via Bcl6b is implicated. As such, the authors should consider assaying the kinetics of MAPK14 phosphorylation (over several time intervals) in response to MAPK7 activation (i.e. constitutively active Map2k5).

Your prediction was correct. Map2k5 transfection increases MAPK7 phosphorylation soon after transfection (Fig 3H), but did not show significant effect on MAPK14 at this point. However, we found that MAPK14 phosphorylation transiently increases after 5-6 days by Map2k5 transfection (Fig 7E). We indicated this in the result section (page 19, line 8 from the bottom). Thank you for your suggestions on this important point.

Minor Comments

4. Regarding the importance of Noxa2 in Nox-1 mediated ROS generation, the authors conclude that "Nox1 expression is the most critical factor in ROS generation" based on the observation that Nox1 transfection alone generates ROS. However, it is possible that endogenous Noxa2 is present in amounts sufficient to support additional Nox1 activity. As such, the authors may consider including data on ROS generation after Nox1 transfection AND Noxa2 shRNA knockdown in Fig 5.

Yes, this is another important point. When we transfected Nox1 cDNA and Noxa2 shRNA, the increase in ROS did occur. This was also included in result section (page 15, line 4 from the bottom)(Fig 5J).

5. For all ROS histogram plots, the authors should indicate the MFIs (or fold-change) of the respective treatment conditions. In addition, it would be ideal if the MFIs from replicate experiments are provided in the form of a bar chart (or equivalent) next to the histogram plots. This will facilitate

comparing the magnitude of change in ROS levels between various experiments (e.g. MAPK14 deletion vs MAPK7 deletion).

We followed your advice and included mean \pm SEM of all flow cytometry data. Because we were not able to find much space in the figure in the main text, we incorporated the data in Appendix Table S1.

However, we are afraid that it is not possible to compare the data in different experiments because we used different settings in some experiments. We described this in the material method section so that readers do not misunderstand this point (page 27, line 2 from the bottom).

6. The authors should also provide ROS data for Nox-1 KO cells in the presence of cytokines or constitutively active Map2k5. This will serve as a reference for the magnitude of ROS generated by NOX1.

We did this experiment and found that Nox1 depletion in cytokine-supplemented cultures or after Map2k5 transfection. Nox1 depletion decreased ROS levels, as we expected (page 6, line 3; page 10, line 7 from the bottom)(Appendix Figs S1B and D).

7. The authors should consider providing quantification (relative to reference gene) for Western blot results.

We quantified all the Western blots, as you suggested. Because we did not have space in the figure, we incorporated the results in Appendix Table S2.

8. The authors may consider including an illustration of the proposed positive feedback model in the last Figure to facilitate comprehension of the study at a glance.

We included a summary figure in the last Figure, as you suggested (Fig 7I).

Thank you for your suggestions, particularly regarding the delayed phosphorylation of MAPK14.

Referee #3:

Thank you for your suggestions to improve our manuscript.

In this study, Morimoto and the colleagues investigate the MAPK-dependent reactive oxygen species (ROS) amplification mechanism, which regulates the self-renewal of mouse spermatogonial stem cells (SSCs). By loss-of-function analysis, the authors showed that specific MAPK proteins are critical for the self-renewal of SSCs and the production of ROS in parallel, suggesting the presence of a feedback mechanism for the amplification of ROS. The following biochemical analysis identified the BCL6B as a mediator for ROS amplification downstream of MAPK signaling. In conclusion, these results suggest that the MAPK14/MAPK7/BCL6B pathway creates a positive feedback loop to promote the self-renewal of SSCs.

Major concerns:

The authors' finding showing the function of specific MAPKs in the self-renewal of SSCs and the production of ROS is novel. The detailed mechanistic analysis would potentially offer novel insights into the growth machinery of SSCs. However, some of the data presented in the manuscript seem to lack coherence and do not appropriately support the hypothesized ROS-amplification mechanism. For example, the authors performed microarray analysis using loss-of-function models and chemical screening analysis to identify the downstream mediators. However, the affected gene profile seems to be quite distinct between these 2 independent analyses. Actually, Bcl6b was identified only in the relatively small-scale chemical screening, suggesting that BCL6B is not a

mediator downstream of MAPK14/7, as the authors claim. This inconsistency also suggests the possibility that BCL6B is one of the off-targets of chemical inhibitors.

Furthermore, throughout the manuscript, the presented results in RT-PCR and Western blotting lack the quantification data to support the authors' claim. Also, in the FACS analysis, it is difficult to estimate to what degree ROS production is affected by a specific condition in each experiment. For much of the data presented in the manuscript, the effects of a specific condition seem to be quite nominal. On the other hand, the signaling intensity measured by FACS analysis is quite changeable, even in the same biological condition as shown in the presented data (e.g. the signal intensity of the control shown in Fig. 6B, C), which suggests that there is considerable variation between experiments. Thus, the authors should present replicated data in each FACS experiment as described in the materials and methods. Moreover, additional measurements of ROS production by other methods would reinforce the authors' claim.

We repeated the FACS experiments three times and the statistical analysis was performed. The reason why the control signals were different in Figs 6B and C was they were taken at different settings. However, we used the same setting in the same experimental comparison. We described this in the material method section so that readers do not misunderstand this point (page 27, line 2 from the bottom)

We chose to use flow cytometry (based on CellROX Deep Red) to measure ROS levels because this allowed detection of intracellular ROS. However, another popular method based on L-012 can only detect extracellular ROS. It is also suggested to reflect cytochrome P450 enzymes (Rezende et al., Free Radic Biol Med 2017; 102:57). AmplexRed is another popular reagent, but this also detects extracellular ROS.

Although H2DCFDA is an alternative method for detecting intracellular ROS, CellROX Deep Red is considered to be superior to H2DCFDA according to manufacturers' instruction. Because we wanted to detect intracellular ROS and how much is produced

by NOX genes, we used flow cytometry and CellROX Deep Red. We also do not have to prepare smaller numbers of cells because L-012 assay requires $1 \times 10^4 - 1 \times 10^5$ cells. These are the reasons why we used this method.

Finally, the authors claim that the positive feedback loop created by MAPK14/MPAK7/BLC6B signaling induces ROS amplification, which promotes SSC self-renewal. However, the presented results do not adequately support this scenario. To confirm the positive feedback loop, the authors should examine whether p-MAPK14 and p-MPAK7 are disrupted in Bcl6b KO SSC/GS cells by the feedback effect.

We carried out the experiment and found that Bcl6b KD decreases the phosphorylation of both MAPK14 and MAPK7. We included the data in result section (page 19, line 5 from the bottom)(Fig 7F).

Additionally, transcriptomic analysis of Bcl6b KO GS cells should be done to assess whether a subset of genes is commonly affected in Bcl6b KO GS cells, Mapk14f/f GS cells, Mapk7f/f GS cells, and Nox1 KD GS cells.

We carried out RNA sequencing and included the data (page 16, line 6)(Appendix Fig S4). However, it was not possible to find commonly affected genes. We speculate that this may be because of the relatively low levels of target gene expression, which is a problem of RNA sequence analysis. Indeed, when we analyzed Nox1 data in RNA sequence, we were not able to find significant difference. As we wrote in the original manuscript, Nox1 expression level is very low despite its importance. However, real-time PCR analysis showed significant downregulation. Therefore, we came to think that capturing these genes may not be easy using microarray or RNA sequencing. Considering that similar experience was also reported (Mehta et al., Cancer Res 2016;76: 7151), it is probably possible that RNA sequencing is not always perfect to detect such subtle changes. We cited this reference in the text to explain our thoughts about this result (page 16, line 9).

Other specific comments:

Fig. 1A

The authors claim that "Mapk11, Mapk12, and Mapk14 are strongly expressed in GS cells" in the manuscript (page 7, line 5). However, the presented data do not support this notion. To clarify how strongly each Mapk gene is expressed in GSCs, the authors should quantify the expression level in several other tissues by real-time PCR.

We carried out real-time PCR and included the data (page 6, line 6)(Appendix Fig S2).

Fig. 1D

The description of the FACS plot is confusing. The authors should describe which colored FACS plot shows the specific condition as shown in Fig. 1E.

We are sorry for confusion. We indicated that the black lines indicate controls.

Fig. 1D, E

To confirm the quantification results by FACS, the authors should present at least two to four FACS replicates as described in the materials and methods.

We carried out three separate experiments. We included the mean fluorescence intensity in all charts.

Fig2.H

The description of the FACS plot is confusing. The authors should describe which colored FACS plot shows the specific condition as shown in Fig. 1E.

To confirm the quantification results by FACS, the authors should present

at least two to four FACS replicates as described in the materials and methods.

We indicated that the black lines indicate controls. We carried out three separate experiments. We included the mean fluorescence intensity for all charts.

Fig2.I

The authors should quantify the image of the Western blot bands. To confirm the regulation of Nox1 by Mapk14, they should also perform real-time PCR.

We are sorry for confusion. The original figure was RT-PCR. We quantified the data by real-time PCR (Fig 2G).

Fig.3E

The description of the FACS plot is confusing. The authors should describe which colored FACS plot shows the specific condition as shown in Fig. 1E.

To confirm the quantification results by FACS, the authors should present at least two to four FACS replicates as described in the materials and methods.

We indicated that the black lines indicate controls. We carried out three separate experiments. We included the mean fluorescence intensity for all charts.

Fig.3F-J

The authors should quantify the band image of the Western analysis. The molecular weight of the specific protein band should be shown in each panel.

We quantified the band, as you suggested (Appendix Table S2). Molecular weight of the band was also indicated in all Western blots.

Fig.3H, I

The band pattern of p-MAPK14 seems to be different between each panel, which is confusing. The cutting of the images should be done in the same manner.

We are sorry for confusion. We repeated the experiment and included a new figure.

Fig.4

The authors should examine whether ROS production is affected in the Mapk7 KO GS cells as is the case in the Mapk14 KO GS cells.

We did this experiment and found that ROS levels are significantly downregulated in Mapk7 KO GS cells (page 14, line 2)(Fig 4F).

Fig.4H

The quantification of the gene expression level should be done by real-time PCR.

We did the experiment and included the data (Fig 4G).

Fig.5B, C

The authors should do the real-time PCR analysis to quantify the gene expression level.

We did the experiment and included the data (Fig 5B, C).

Fig.5D, F

The description of the FACS plot is confusing. The authors should describe which colored FACS plot shows the specific condition as shown in Fig. 1E.

To confirm the quantification results by FACS, the authors should present

at least two to four FACS replicates as described in the materials and methods.

We indicated that the black lines indicate controls. We carried out three separate experiments. We included the mean fluorescence intensity for all charts.

Fig.5F

Here, the authors claim that Nox1 transfection alone is sufficient to generate ROS. To this reviewer, however, the knockdown experiment seems to suggest that Noxa2 plays a critical function in ROS production. As the following overexpression experiment is done in the presence of endogenous Noxa2, it is not clear how Nox1 alone is sufficient to generate ROS. To confirm their claim, the authors should assess whether Nox1 OE can rescue the reduction of ROS generation even in the Noxa2 KD condition. They should also overexpress Noxa2 and examine the effect on ROS production and SSC self-renewal.

Yes, this was a reasonable question. With your suggestion, we carried out real-time PCR and found that Nox1 and Noxa2 expression and regulation is not exactly the same. We transfected Nox1 cDNA and Noxa2 shRNA and found that ROS levels did increase (page 15, line 4 from the bottom)(Fig 5J). We also transfected Noxa2 cDNA. However, there was no increase in ROS production or proliferation (page 15, line 7 from the bottom)(Figs 5I and F).

Fig.5H

The authors should quantify both Noxa1 and Noxa2 levels by real-time PCR analysis.

We did the experiment and found that only Nox1 was upregulated (I suppose that your meant Nox1 instead of Noxa1)(Fig 5D).

Fig.6A

As described above, the presented data and related descriptions do not show the MAPK14/MAPK7/BCL6B pathway as the authors claim. The authors should explain the inconsistency between the microarray analysis and the chemical screening. They should also exclude the possibility that these results are caused by off-target effects.

The quantification of gene expression should be done by real-time PCR.

We think that the sensitivity of microarray is not high enough to detect Nox1 expression level. We also carried out RNA sequence analysis and still could not find significant changes. We think that this was because of the low sensitivity of the assays. We wrote this possibility in the result section (page 16, line 9).

We also evaluated the off-target effects by using another set of inhibitors and included the data (page 16 line 5 from the bottom)(Fig EV4B). Thank you for pointing this out. We were able to reduce the number of candidate genes. Although Bcl6b was confirmed, Sohlh1 did not respond as we expected.

Fig.6B, C, Appendix Fig S4

The knockdown efficiency of Bcl6b and other estimated factors should be confirmed by real-time PCR.

The description of the FACS plot is confusing. The authors should describe which colored FACS plot shows the specific condition as shown in Fig. 1E.

To confirm the quantification results by FACS, the authors should present at least two to four FACS replicates as described in the materials and methods.

We estimated the knockdown efficiency by real-time PCR, as you suggested (Appendix Fig S3).

We indicated that the black lines indicate controls. We carried out at three separate experiments. We included the mean fluorescence intensity for all charts.

Fig.7B, D

In Fig. 7A, the signal intensity of BCL6B in the overexpression conditions seems to be noticeably higher than that of the control, which is inconsistent with the modest activation of BCL6B by Map2k5 as shown in Fig 7A. In Fig 7D, however, the BCL6B signal of the control seems to be higher than the control signal in Fig. 7B, which is confusing. The authors should present the image in low-power field.

Additionally, to confirm the result, the authors should perform real-time PCR and Western blotting.

We added images in low-power field, as you suggested (Appendix Fig S5A).

We did real-time PCR and Western blot to confirm that there are no big changes in RNA and protein levels of Bcl6b gene (page 19, line 10 from the bottom)(Appendix Figs S5B and S5C).

Fig.7C

The description of the FACS plot is confusing. The authors should describe which colored FACS plot shows the specific condition as shown in Fig. 1E.

To confirm the quantification results by FACS, the authors should present at least two to four FACS replicates as described in the materials and methods.

We indicated that the black lines indicate controls. We carried out three separate experiments. We included the mean fluorescence intensity for all charts.

Fig.7E, F

The representative images seem to show that BCL6B is not expressed in Mapk14 KO and Mapk7 KO GS cells. However, the authors quantified the levels of BCL6B in the nucleus in Mapk14 KO and Mapk7 KO GS cells, which is confusing. The authors should present the image in low-power

field.

Furthermore, to confirm the result, the authors should perform real-time PCR and Western blotting.

Although the signal is not very strong, there are BCL6B protein in both Fig 7E and F. As you suggested, we incorporated images in low-power field (Appendix Fig S5A). We also carried out real-time PCR (Appendix Fig S5B) and Western blot (Appendix Fig S5C), but did not find significant changes.

Thank you for your suggestions, particularly regarding the delayed phosphorylation of MAPK14 and real-time PCR analysis.

Referee #3:

While the authors have addressed several specific concerns, especially the quantification issues in each experiment to fulfill this reviewer's claims, they have not clarified the reviewer's major concern, unfortunately. Specifically, for the hypothesized ROS-amplification pathway, MAPK14/7/BCL6B is still not adequately demonstrated in the manuscript, making it difficult to ascertain that this pathway is critical for ROS amplification and SSC self-renewal. This reviewer pointed out the inconsistency between two independent experiments (i.e. microarray in the Mapk7/14 mutant and gene expression changes caused by inhibition of MAPK via small molecules) and asked the authors to clarify whether BCL6B, which was identified only in the chemical assay, is truly a mediator that creates the feedback loop downstream of MAPK7/14, as they claim. The authors, however, did not present any additional data or a coherent explanation for that matter to support their view. If they see that the inconsistency comes mainly from the low detection level of the microarray or the RNA sequence like in the case of Nox1, they should at least examine whether Bcl6b, and other subset of genes identified in the chemical assay (Fig EV4), was also commonly affected in the Mapk7/14 mutant, the Nox1 mutant, and the Bcl6b1 mutant by real-time qPCR, as this reviewer commented. Related to this point, the RNA-sequence data does not seem to be consistent with the microarray data-the top 50 genes screened in RNA sequencing were very different from those of the microarray.

Referee #3:

While the authors have addressed several specific concerns, especially the quantification issues in each experiment to fulfill this reviewer's claims, they have not clarified the reviewer's major concern, unfortunately. Specifically, for the hypothesized ROS-amplification pathway, MAPK14/7/BCL6B is still not adequately demonstrated in the manuscript, making it difficult to ascertain that this pathway is critical for ROS amplification and SSC self-renewal. This reviewer pointed out the inconsistency between two independent experiments (i.e. microarray in the *Mapk7/14* mutant and gene expression changes caused by inhibition of MAPK via small molecules) and asked the authors to clarify whether BCL6B, which was identified only in the chemical assay, is truly a mediator that creates the feedback loop downstream of MAPK7/14, as they claim. The authors, however, did not present any additional data or a coherent explanation for that matter to support their view. If they see that the inconsistency comes mainly from the low detection level of the microarray or the RNA sequence like in the case of *Nox1*, they should at least examine whether *Bcl6b*, and other subset of genes identified in the chemical assay (Fig EV4), was also commonly affected in the *Mapk7/14* mutant, the *Nox1* mutant, and the *Bcl6b1* mutant by real-time qPCR, as this reviewer commented. Related to this point, the RNA-sequence data does not seem to be consistent with the microarray data-the top 50 genes screened in RNA sequencing were very different from those of the microarray.

With your suggestion, we carried out real-time PCR to examine the expression of candidate genes found by chemical studies. As we expected, *Bcl6b* was downregulated in *Mapk14* KO, *Mapk7* KO and *Nox1* KD GS cells (Fig EV4C). Although the impact was relatively weak compared with the chemical studies, this analysis also suggested that *Bcl6b* is the strongest candidate.

Referee #3:

To this reviewer, the MAPK7/14/BCL6b pathway for ROS amplification is still unclear. Since the down-regulation in the Bcl6b expression by Mapk7/14 deletion is limited, i.e. around 70-80% of the wild type level in 6 replicates (Fig. EV4C), this result rather suggests that the role of MAPK7/14 in Bcl6b1 activation for ROS-amplification is not a major pathway as the authors claim. It is also quite possible that other (known or unknown) factors than Mapk7/14 affected by small molecules are critical for the activation of Bcl6b1. Indeed, small molecules show a relatively strong negative effect on many other factors whose deficiency causes defective spermatogenesis (Fig. EV4A), while the effect on these factors is relatively small in Mapk7/14 mutant (Fig. EV4C).

Additionally, the authors should explain the inconsistent result between RNA-seq and microarray, the quite different outcome of screened genes, which this reviewer pointed out in the previous review. They should also present the raw signal intensity of gene sets shown in Fig. EV4C to support the validity and their claim.

March 11, 2019

RE: Life Science Alliance Manuscript #LSA-2019-00374-T

Dr. Takashi Shinohara
Kyoto University
Molecular Genetics
Yoshida Konoe
Sakyo
Kyoto, Kyoto 606-8501
Japan

Dear Dr. Shinohara,

Thank you for transferring your revised manuscript entitled "ROS amplification drives mouse spermatogonial stem cell self-renewal" to Life Science Alliance. Your manuscript was reviewed several times at another journal before, and the editors transferred those reports to us with your permission.

The reviewer who evaluated the revised versions of your work thought that it remains unclear whether MAPK7/14/Bcl6b1 activation for ROS-amplification is a major pathway, pointing out that there may be MAPK7/14-independent pathways leading to Bcl6b1 activation. We think this concern can get addressed by text changes, leaving room for alternative explanations for the observations made. The remaining concern regarding inconsistency between RNA-seq and microarray data should get addressed by further discussion and the request for raw signal intensity of gene sets for current Fig. EV4C should get addressed. We would then be happy to accept such a further revised manuscript for publication here.

Please also address the following editorial points when preparing the final version of your work:

- please note that at LSA we only have supplementary figures and tables (no EV figures nor Appendix files). I'd appreciate if you could revise your manuscript accordingly.
- please note that current figure EV5 is missing the panel descriptors ('A', 'B'...)
- please upload all figure files as individual ones
- please note that currently a callout to appendix table S4 is missing from the text
- please link your ORCID iD to your profile in our submission system, you should have received an email with instructions on how to do so

A. FINAL FILES:

B. MANUSCRIPT ORGANIZATION AND FORMATTING:

March 18, 2019

RE: Life Science Alliance Manuscript #LSA-2019-00374-TR

Dr. Takashi Shinohara
Kyoto University
Molecular Genetics
Yoshida Konoe
Sakyo
Kyoto, Kyoto 606-8501
Japan

Dear Dr. Shinohara,

Thank you for submitting your Research Article entitled "ROS amplification drives mouse spermatogonial stem cell self-renewal". I appreciate the introduced changes and it is a pleasure to let you know that your manuscript is now accepted for publication in Life Science Alliance. Congratulations on this interesting work.

DISTRIBUTION OF MATERIALS:

Again, congratulations on a very nice paper. I hope you found the review process to be constructive and are pleased with how the manuscript was handled editorially. We look forward to future exciting